complexity/behaviour/mathematical modelling

dynamical systems, evolutionary game theory, linguistic convergence and divergence, replicator–mutator equation, sociolinguistics

**Author for correspondence:**
Henri Kauhanen
e-mail: henri.kauhanen@uni-konstanz.de

# Replicator–mutator dynamics of linguistic convergence and divergence

## Henri Kauhanen

Zukunftskolleg, University of Konstanz, Konstanz, Germany

HK, 0000-0002-6743-5100

People tend to align their use of language to the linguistic behaviour of their own ingroup and to simultaneously diverge from the language use of outgroups. This paper proposes to model this phenomenon of sociolinguistic identity maintenance as an evolutionary game in which individuals play the field and the dynamics are supplied by a multi-population extension of the replicator–mutator equation. Using linearization, the stabilities of all dynamic equilibria of the game in its fully symmetric two-population special case are found. The model is then applied to an empirical test case from adolescent sociolinguistic behaviour. It is found that the empirically attested population state corresponds to one of a number of stable equilibria of the game under an independently plausible value of a parameter controlling the rate of linguistic mutations. An asymmetric three-population extension of the game, explored with numerical solution methods, furthermore predicts to which specific equilibrium the system converges.

## 1. Introduction

Most mathematical models of linguistic variation and change fall into one of two classes: (i) those that study homogeneous populations of speakers in frameworks that admit an analytical exploration of the model's dynamics [1–3], and (ii) those that aim to elucidate complex interactions of language learning and use in social networks through agent-based simulations [4–6]. The former class is usually characterized by the use of continuous-time dynamics in infinite populations; such significant idealizing assumptions often limit the range of empirical application of these models. On the other hand, models in the latter class are characterized by more microscopic attention to detail, and perhaps better empirical applicability, but at the expense of increasingly complex parameter spaces and a concomitant loss of analytical tractability outside of special cases (although sometimes considerable progress can be made by considering suitable

reductions of full models, such as continuous-time limits; see e.g. [7]). This motivates the search for a modelling paradigm that falls somewhere between the two extremes.

Outside of modelling, the use of macrosocial categories such as socioeconomic class and gender is well established within the broad field of sociolinguistics [8]. This basic insight remains underexploited in the literature on mathematical models of linguistic variation and change, however, where attention ordinarily focuses on individual speakers and their interactions. Macroscopic multi-population modelling is not uncommon in evolutionary biology [9], epidemiology [10] and economics [11], where it often supplies analytical results which approximate empirical reality and provide useful Archimedean points which may assist in the analysis of more complex microscopic models. It may be expected to perform a similar role in elucidating processes of variation and change in the domain of language.

In this paper, I explore a macroscopic model of language dynamics whose inspiration is taken from a well-established empirical observation: that speakers tend to converge in their use of language to that of their immediate social group, and diverge from language use in other, possibly polarly opposed, social groups [12–15]. These processes together contribute to the construction and maintenance of sociolinguistic identity: as different social groups diverge in their use of language, those contrasting linguistic behaviours take on the function of indexing group identity. The model is thus put forward both as a mathematical formalization of linguistic identity dynamics, mediated through convergence and divergence, and as an exploration of what sorts of benefits may be reaped from the study of structured populations at a high level of abstraction. The model assumes a distinction between a number of social groups but treats individual speakers within each group identically—putting, in the spirit of the earliest sociolinguistic traditions, the community before the individual [16, p. 7]. Indeed, the model assumes continuous-time dynamics operating in a compartmentalized infinite population with fully random mixing within and across compartments. Nevertheless, it is rich enough to predict an empirically attested sociolinguistic pattern and to suggest additional predictions for further empirical testing.

A lively debate now exists about how many and what ingredients are minimally needed in a mathematical model of linguistic variation and change, with proposals ranging from neutral or near-neutral models [17–19] to models incorporating intrinsic biases acting on competing linguistic variants [4,20]. The crucial issue is whether simple frequency matching in an environment characterized by various accidental asymmetries and stochastic noise is sufficient to give rise to classical patterns of variation and change, such as the S-curve [21], or whether stronger assumptions about deterministic biases are needed (see [22] for pertinent discussion and a useful taxonomy of modelling approaches). The results presented in this paper suggest that classical patterns of variation and change can arise from an intuitively plausible source, but nevertheless one that has not received much attention in the modelling literature—the opposing forces arising from the dynamics of subpopulations which lie, in one sense or another, in an antagonistic relation to each other. In such a setting, said patterns may arise even in the absence of directed biases acting on the linguistic variants themselves, that is to say, variation and change may in these cases be largely externally, more particularly socially, governed.

The main contribution of the present paper, however, is in showing that linguistic identity dynamics, and the ensuing social stratification of linguistic variants, can be usefully modelled using tools from the mathematical theory of evolutionary games [23,24]. A formal investigation of this model reveals that the 'convergence–divergence game', as I shall call it below, has a number of stable equilibria depending on a combination of global parameters, namely, parameters describing the strength of ingroup social convergence and outgroup divergence on the one hand, and parameters describing the linguistic mutation (innovation or mistransmission) dynamics of competing variants, on the other. Empirically, one of these stable equilibria is found to correspond to data on sociolinguistic stratification in adolescents [25] for an independently estimated value of the relevant mutation rate parameter. This result, derived from a simple instance of the game which is analytically tractable, lends a degree of empirical plausibility to the mathematical model. Consideration of the minutiae of real-life linguistic and social dynamics requires, however, certain relaxations of idealizing assumptions in the simple model, motivating the numerical exploration of a somewhat more complex instance of the model, as will be discussed at length below.

A number of applications of evolutionary game-theoretic (EGT) modelling are now available in the literature on language dynamics [26–35]. These will be briefly discussed in §2 in order to contextualize the contribution of the present paper. In §3, the convergence–divergence game is defined on a general, abstract level. In the language of EGT, the game incorporates nonlinear fitness and thus falls among the class of the so-called playing-the-field games [36]. The practical consequence of this is that the mathematical analysis of the game is non-trivial, as existing results for games with linear fitness (matrix games; [37]) are not available. Nevertheless, in §4, I show that the dynamic equilibria of the nonlinear convergence–divergence game can be solved in the case of two strategies competing in two populations,

if the effects of social alignment and mutation are symmetric in a sense to be discussed below. Armed with these analytical results, in §5 I proceed to compare one of the stable equilibria with Eckert's [25] data on the backing of the STRUT vowel /ʌ/ in two contrasting but interacting adolescent social groups, 'jocks' and 'burnouts'. Crucially, the mutation rate can be estimated in this case from available literature on the confusability of speech sounds [38]. Given this estimate, it is found that the sociolinguistic data are predicted by the formal game, in the sense that the empirical data point falls in a phase of the game in which a number of relevant stable equilibria are available. In §6, I explore an extension of the game for multiple (more than two) populations, taking note of the crucial role that the population of adults plays in the empirical situation. Although analytical tractability is lost, numerical solutions of the multi-population model show that the model again aligns with empirical data. Importantly, the extension explains why one rather than another equilibrium is attained in this particular situation. These findings have implications for the mathematical modelling of language dynamics as well as the empirical study of sociolinguistics; they will be discussed in §7.

# 2. Evolutionary games and language dynamics

At the most abstract level, an evolutionary game is a formal representation of frequency-dependent selection (and possibly mutation) in a population of interacting players. Each player has access to some set of strategies, and upon meeting other players in a sequence of contests, chooses one of these. The outcome of a contest is a pay-off to each interacting party. Classically, pay-offs are assumed to be fixed functions of the strategies involved in the contest, and can be represented in a matrix if contests are pairwise [37]. In such a game, the pay-offs determine the fitness of each strategy and, together with the game's dynamic (see below) also the evolutionary fate of the population, such as what strategies, or perhaps mixes of strategies, are stable.

Since its emergence in the 1970s [36,39,40], EGT has reached a level of maturity in the fields of mathematical biology and economics. As several authors have noted, however, this framework is also applicable to the modelling of language dynamics, if its ingredients are suitably interpreted [26–35]. In general, it is possible to equate strategies with linguistic variants, i.e. the values of a linguistic variable [31], or with entire grammars [26]. Similarly, contests can be interpreted either as events of linguistic interaction [34], or as an abstract cumulative representation of the extended process of language acquisition in childhood [28]. The interpretation of pay-offs and fitness also varies from application to application. In [31], for example, the fitness of strategies (competing form–meaning mappings) results from a combination of communicative success and formal linguistic economy. In [33], on the other hand, strategies constitute competing stress placements on lexical items (the 'players' of the game) and fitness results from eurhythmicity (or lack thereof) at the level of entire phrases. It should be noted that the notion of fitness is by no means new to linguistics, but rather, the above interpretations are related in obvious ways to traditional accounts of the differential adaptedness of linguistic variants, ranging from communicative function and contrast maintenance [41,42] through prestige and other kinds of social biases [20,43] to parsing advantage [44,45] and economy of computation, perception or articulation [46,47].

One advantage of EGT-based modelling over other approaches, such as agent-based simulations, is that analytical results can often be obtained if the assumptions entering the game's definition are suitably abstract. This renders EGT models transparent [33], as the complete phase space of the model can be succinctly characterized, along with any possible bifurcations the system may undergo in response to variation in its control parameters. Even when analytical tractability is lost, macro-level dynamical-systems modelling has the advantage that the numerical solution of the relevant differential equations is considerably less resource-intensive than the computer simulation of corresponding agent-based models. This fact is exploited in §6, when the convergence–divergence game is generalized for more than two interacting populations.

In addition to specifying how pay-offs result from contests, an application of EGT needs to specify a dynamic for evolving the population; this encodes assumptions about how strategies replicate in the population and, possibly, how new strategies may emerge. A canonical dynamic is the continuous-time replicator equation [23,39], which for two strategies (the case discussed in the present paper) reads

$$\dot{x} = x(1-x)[f(x) - \tilde{f}(x)]. \tag{2.1}$$

Here, $x$ ($0 \le x \le 1$) represents the frequency of one of the strategies in the population, $\dot{x}$ is its time derivative, $f(x)$ its fitness, and $\tilde{f}(x)$ the fitness of the competing strategy (the frequency of the competing strategy is, of course, $1-x$). In very general terms, the shape of equation (2.1) implies that the usage of a

strategy increases (decreases) whenever its fitness is greater (lower) than that of its competitor. The replicator equation can thus serve as a basic formalization of Darwinian selection [37], but crucially for applications in cultural evolution, it can also be interpreted as describing the differential replication of non-biologically transmitted variants [33,34]. As a continuous-time differential equation, it formally describes the evolution of strategies in a hypothetical infinite, fully mixing population. However, and again crucially from the point of view of applications in cultural evolution, the equation can be derived as the continuous limit of a stochastic imitation dynamic in which players imitate each others' strategies in relation to their fitnesses [24].

It is important to stress that the fitness terms $f(x)$ and $\tilde{f}(x)$ in equation (2.1) are functions of the population state $x$. Hence, the fitness of a strategy is never static (except in mathematically degenerate, trivial games), but rather evolves as the population evolves. This fact guarantees that the dynamics can display a number of interesting behaviours ranging from simple point attractors to cycles and bifurcations [23]. But it also highlights the philosophically important observation that in the dynamics of language, too, the evolutionary success of a strategy (such as a particular variant of a sociolinguistic variable) must depend on what other strategies are in use in the population, and at what frequencies. These facts will be discussed in more depth in the following section.

# 3. A game of convergence and divergence

A considerable body of evidence suggests that people tend to accommodate their use of language to the linguistic behaviour of those they feel close to or view in positive terms, and to distance their use of language from the behaviour of those who are distant or negatively assessed [12–15]. These findings have been systematized under the umbrella of communication accommodation theory [13], which identifies two principal mechanisms of accommodation: convergence and divergence. Broadly speaking, the goal of linguistically convergent behaviour is to decrease the social distance between the interlocuting parties, whereas divergent behaviour (such as choosing different pronunciations, different registers, different speech rates) tends to increase that social distance and to contribute to the upkeep of socially defined identities. The processes of linguistic convergence and divergence may be to some extent automatic [48] and, even though possibly deeply rooted, may not always be conscious. For example, a study exploring the extent to which speakers of New Zealand English accommodated to a speaker of Australian English who was instructed to either insult or flatter the former [14] presented nuanced findings. Even though the experimental condition (insult versus flattery) did not predict the degree of convergence or divergence (measured in terms of the similarity or dissimilarity of vowel sounds), the results of an implicit association task [49] showed that those New Zealand speakers who scored high for implicit pro-Australian biases also attested significantly more linguistic convergence to the Australian speaker than those who scored lower for such biases.

A number of possible interpretations of fitness in the context of linguistic variation and change were discussed in §2. In most of these cases, fitness is intrinsic to linguistic strategies, in the sense that it is determined by the functional, formal or, broadly speaking, 'purely linguistic' properties of those strategies. For example, in the game explored in [33], the fitness of a particular stress placement on polysyllables results from the configuration of the rest of the linguistic system, namely, how frequent monosyllables are in the language (which turns out to determine phrase-level rhythmicity when polysyllabic and monosyllabic words combine to form phrases). Clearly, however, the fitness of a variant may also be socially accrued. In particular, the dual processes of linguistic convergence and divergence may be (partly) responsible for fitness: for me, the fitness of a strategy increases the more those in my own social group (my ingroup) use this strategy but also decreases the more those in other groups (outgroups) use it. I shall call this sort of fitness *extrinsic*.

Although previous studies have modelled accommodation to community behaviour by way of various mechanisms of frequency matching [6,18,19,22,50], extrinsic fitness has rarely, if ever, been included as an explicit macro-level term in mathematical models of language dynamics. The proposal here is to consider, initially, two population substrata, which could be thought to correspond to two socioeconomic classes, two subgroups of an age cohort, two age cohorts in a single community, or in general any two subpopulations of a speech community whose members are inclined to converge toward the speech patterns of those in their own subpopulation but diverge from the usage of those in the other subpopulation.[1] The crucial assumption is that the two groups be unambiguously

---

[1]In the real world, processes of convergence and divergence are much more nuanced than this simple set-up would suggest. These complexities will be further discussed in §7.

distinguishable in terms of non-linguistic markers, so that the group membership of most speakers in the community is not in doubt for members of either subcommunity. This can normally be assumed to be the case, as socially salient group distinctions tend to be indicated by various, visible extra-linguistic semiotic practices, such as choice of clothing and fashion [51].

Generalizing directly from the discussion in §2, let $x_1$ and $x_2$ denote the frequency of use of a strategy in two interacting populations; for symbolic ease, these are often collected in a vector $\mathbf{x} = (x_1, x_2)$ in what follows. Intuitively, the fitness of this strategy in the two populations should depend, firstly, on how frequently the two populations interact. Secondly, it should depend on how strong the tendencies to converge to ingroup behaviour and to diverge from outgroup behaviour are. To formalize these ideas, let

$$P = \begin{pmatrix} p_{11} & p_{12} \\ p_{21} & p_{22} \end{pmatrix} \tag{3.1}$$

be a row-stochastic matrix of interaction rates, such that cell $p_{ij}$ gives the probability of an individual from population $i$ interacting with an individual from population $j$. Similarly, let

$$A = \begin{pmatrix} a_{11} & a_{12} \\ a_{21} & a_{22} \end{pmatrix} \tag{3.2}$$

be a matrix of real numbers. Here, $a_{11}$ and $a_{22}$ supply, for want of a better term, the 'strength' of convergence to ingroup norm for the two populations, while $a_{12}$ and $a_{21}$ supply the concomitant strengths of divergence from outgroup norm. The matrix $A$ is not required to be row-stochastic, but in view of the intended interpretation, we require its elements to be positive.[2]

The fitness of one of two competing strategies in the two populations can now be defined as follows:

$$\left. \begin{array}{l} f_1(\mathbf{x}) = p_{11}a_{11}x_1 + p_{12}a_{12}(1 - x_2)x_1 \\ f_2(\mathbf{x}) = p_{22}a_{22}x_2 + p_{21}a_{21}(1 - x_1)x_2 \end{array} \right\} \tag{3.3}$$
and

(to be explained in detail shortly). For the fitness of the competing strategy we similarly have

$$\left. \begin{array}{l} \tilde{f}_1(\mathbf{x}) = p_{11}\tilde{a}_{11}(1 - x_1) + p_{12}\tilde{a}_{12}x_2(1 - x_1) \\ \tilde{f}_2(\mathbf{x}) = p_{22}\tilde{a}_{22}(1 - x_2) + p_{21}\tilde{a}_{21}x_1(1 - x_2) \end{array} \right\} \tag{3.4}$$
and

for another convergence/divergence matrix $\tilde{A} = [\tilde{a}_{ij}]$. In each of these equations, the first term on the right-hand side accounts for intra-group interactions, the second for inter-group interactions. Thus, from the first term, the fitness of the strategy increases the more intra-group interactions take place, the more important convergence to ingroup norm is judged to be, and the more the strategy is employed in the ingroup. Conversely, from the second term, the fitness of the strategy increases the more inter-group interactions take place, the more important divergence from outgroup norm is judged to be, and the more the strategy is *not* used in the outgroup. The second term also includes the ingroup frequency as a factor. This potentially puzzling choice can be explained as follows: the status of the strategy *qua* identity marker depends on how widespread it is in the ingroup; a strategy can demarcate one group from another only to the extent it is an established norm in the ingroup. The practical consequence of this choice is that fitness is a nonlinear function of the population state $\mathbf{x} = (x_1, x_2)$. Consequently, the model belongs to the class of playing-the-field games, and formal results established for matrix (linear fitness) games do not necessarily apply.

To close the definition of the convergence–divergence game, we need to consider a dynamic for evolving the two populations. Although there is precedent for using the replicator equation (e.g. [33]), this has the downside that pure replicator dynamics lack a mutation term and thus can account for neither the emergence of strategies not already present in the population, nor for the effects of mistransmission. To include these aspects in the game, I consider the replicator–mutator equation [52]. For two strategies in two populations, it reads (see appendix A.1):

$$\left. \begin{array}{l} \dot{x}_1 = (\tilde{x}_1 - \tilde{m}_1)x_1 f_1(\mathbf{x}) - (x_1 - m_1)\tilde{x}_1 \tilde{f}_1(\mathbf{x}) \\ \dot{x}_2 = (\tilde{x}_2 - \tilde{m}_2)x_2 f_2(\mathbf{x}) - (x_2 - m_2)\tilde{x}_2 \tilde{f}_2(\mathbf{x}), \end{array} \right\} \tag{3.5}$$
and

where I write $\tilde{x}_i = 1 - x_i$ for convenience, and $\tilde{m}_i$ ($m_i$) is the rate with which the first (second) strategy mutates into the second (first) in population $i$. The replicator dynamics are recovered as the special

[2]Unless otherwise noted, I assume throughout that the matrices $P$ and $A$ do not contain zeroes. A zero element means that either interaction or convergence/divergence vanishes completely; these degenerate cases are of technical mathematical interest only.

case $\tilde{m}_1 = m_1 = \tilde{m}_2 = m_2 = 0$. The mutation rates can also usefully be collected in matrices:

$$M_1 = \begin{pmatrix} 1 - \tilde{m}_1 & \tilde{m}_1 \\ m_1 & 1 - m_1 \end{pmatrix} \quad \text{and} \quad M_2 = \begin{pmatrix} 1 - \tilde{m}_2 & \tilde{m}_2 \\ m_2 & 1 - m_2 \end{pmatrix}. \tag{3.6}$$

We note that the phase space of the system is the unit square $[0, 1]^2 = \{(x_1, x_2) \in \mathbf{R}^2 : 0 \leq x_1, x_2 \leq 1\}$.

The inclusion of the mutation rate parameters in equation (3.5) is intended to model the fact that linguistic 'mutations'—misarticulations, misperceptions, misparsings, external borrowings, proper innovations, etc.—occur in language continuously, even though much of the time they fall below some threshold and fail to actuate a change [53,54]. From a conceptual point of view, it is important to note that while mutations must eventually show up in linguistic production, their distal causes may be perceptual, and nothing about equation (3.5) precludes mutations from arising from the listener (cf. [46]). The population-level effects of misproduction and misparsing (or failure to apply reconstructive rules, as in [46]) may very well be identical, and the simple model here studied makes no ontological distinction between the two sources of mutation.

Before proceeding to a detailed study of the properties of this game, I point out that notation can be further simplified if the interaction and convergence/divergence matrices are folded together. Writing $S$ for the elementwise product of $P$ and $A$, and $\tilde{S}$ for that of $P$ and $\tilde{A}$ (i.e. $s_{ij} = p_{ij}a_{ij}$ and $\tilde{s}_{ij} = p_{ij}\tilde{a}_{ij}$), and taking advantage of the convention $\tilde{x}_i = 1 - x_i$, we have

$$\left. \begin{aligned} f_1(\mathbf{x}) &= s_{11}x_1 + s_{12}\tilde{x}_2 x_1, \\ f_2(\mathbf{x}) &= s_{22}x_2 + s_{21}\tilde{x}_1 x_2, \\ \tilde{f}_1(\mathbf{x}) &= \tilde{s}_{11}\tilde{x}_1 + \tilde{s}_{12}x_2 \tilde{x}_1 \\ \tilde{f}_2(\mathbf{x}) &= \tilde{s}_{22}\tilde{x}_2 + \tilde{s}_{21}x_1 \tilde{x}_2 \end{aligned} \right\} \tag{3.7}$$

and

for the fitnesses. In what follows, I will refer to $S$ and $\tilde{S}$ as *social alignment* matrices and to their elements as social alignment parameters.

Although not much can be said about the behaviour of this system without imposing further restrictions on the social alignment and mutation rate parameters, the following general result will be useful later.[3]

**Proposition 3.1.** *The number of equilibria of the two-strategy, two-population convergence–divergence game is at least one and at most nine.*

# 4. Symmetry

Rigorous mathematical analysis of replicator–mutator dynamics is challenging, even when fitness is linear [26,27,55,56]. In the nonlinear case here considered, progress can be made if rather strong symmetry conditions are imposed on the social alignment and mutation matrices, thereby reducing the number of free parameters of the system. Given these assumptions about symmetry, it turns out that it is possible to find the game's dynamic equilibria and their stabilities analytically, as well as to delineate how these depend on the model parameters. This allows for a complete specification of the possible behaviours of the system, which will aid us in comparing the model's predictions with empirical data in the sections to follow.

There is some reason to think that the model, as above defined, is in fact too general. Thus consider the fact that we have two mutation matrices $M_1$ and $M_2$ instead of one: this allows for the possibility that the mutation rates of strategies differ from one population to the next. Insofar as linguistic mutations arise from universal principles of (mis)perception, (mis)articulation or (mis)computation [46,47], this generality may be superfluous, or in other words, we may just as well assume $M_1 = M_2$ without much worry. On the other hand, the inclusion of four free parameters in each of the social alignment matrices $S$ and $\tilde{S}$ may seem superfluous. In particular, this allows for the possibility that the effects of convergence and divergence are different in the two populations (this occurs if $s_{11} \neq s_{22}$ and/or $s_{12} \neq s_{21}$ in the case of the first strategy, and similarly with the tilde'ed parameters for the second strategy, i.e. if $S$ and $\tilde{S}$ are not bisymmetric matrices). But in the absence of evidence for such between-population differences, we may assume the populations to be equivalent in this respect and thus impose the restriction that the matrices $S$ and $\tilde{S}$ be bisymmetric. If it is the case that both (i) $M_1 = M_2$ and (ii) $S$ and $\tilde{S}$ are bisymmetric, then the two populations are interchangeable from the point of view of each strategy, and I will call this configuration *population symmetry* in what follows.

---

[3]Proofs of all formal results are collected in appendix A.2.

Conversely, we may imagine a situation in which $m_1 = \tilde{m}_1$ and $m_2 = \tilde{m}_2$, i.e. the matrices $M_1$ and $M_2$ are bisymmetric, so that mutations between the two strategies are equally likely in each direction. If, moreover, $S = \tilde{S}$, each strategy responds to convergence and divergence identically within each population. Whenever (i) $M_1$ and $M_2$ are bisymmetric and (ii) $S = \tilde{S}$, I shall say that the configuration exhibits *strategy symmetry*. Finally, whenever a system exhibits both population symmetry and strategy symmetry—that is to say, whenever (i) $M := M_1 = M_2$, (ii) $S = \tilde{S}$, and (iii) $M$ and $S$ are bisymmetric matrices—I shall say it exhibits *full symmetry*.

Full symmetry leads to a significant reduction in the number of free parameters of the game, thereby facilitating its analysis. Since $S = \tilde{S}$ and the matrix is assumed bisymmetric, the fitnesses (3.7) now read

$$\left.\begin{aligned}
f_1(\mathbf{x}) &= s_{11}x_1 + s_{12}x_1\tilde{x}_2, \\
f_2(\mathbf{x}) &= s_{11}x_2 + s_{12}x_2\tilde{x}_1, \\
\tilde{f}_1(\mathbf{x}) &= s_{11}\tilde{x}_1 + s_{12}\tilde{x}_1x_2 \\
\tilde{f}_2(\mathbf{x}) &= s_{11}\tilde{x}_2 + s_{12}\tilde{x}_2x_1.
\end{aligned}\right\} \tag{4.1}$$

and

Assuming $s_{11} \neq 0$ and writing $\sigma = s_{12}/s_{11}$, (4.1) can be further reduced to the following representation:

$$\left.\begin{aligned}
f_1(\mathbf{x}) &= x_1 + \sigma x_1\tilde{x}_2, \\
f_2(\mathbf{x}) &= x_2 + \sigma x_2\tilde{x}_1, \\
\tilde{f}_1(\mathbf{x}) &= \tilde{x}_1 + \sigma \tilde{x}_1x_2 \\
\tilde{f}_2(\mathbf{x}) &= \tilde{x}_2 + \sigma \tilde{x}_2x_1.
\end{aligned}\right\} \tag{4.2}$$

and

The equivalence of (4.1) and (4.2) is guaranteed by the following result on the topological equivalence of the two dynamical systems:

**Proposition 4.1.** *The games defined by* (4.1) *and* (4.2) *share the same orbits, and in particular the same equilibria, under the dynamic* (3.5).

Writing $\mu = m_1 = \tilde{m}_1 = m_2 = \tilde{m}_2$ for the common mutation parameter, the study of the fully symmetric convergence–divergence game thus reduces to a study of the system

$$\left.\begin{aligned}
\dot{x}_1 &= (\tilde{x}_1 - \mu)x_1f_1(\mathbf{x}) - (x_1 - \mu)\tilde{x}_1\tilde{f}_1(\mathbf{x}) \\
\dot{x}_2 &= (\tilde{x}_2 - \mu)x_2f_2(\mathbf{x}) - (x_2 - \mu)\tilde{x}_2\tilde{f}_2(\mathbf{x})
\end{aligned}\right\} \tag{4.3}$$

and

with fitnesses (4.2).

By proposition 3.1, this game has between one and nine equilibria; in other words, the number of possible long-term outcomes of the population dynamics must fall within these bounds. The mutation rate parameter $\mu$ turns out to play a decisive role in determining the number of equilibria and their stabilities. Using linearization [57], it is possible to establish that this parameter has three critical values, $\mu_1$, $\mu_2$ and $\mu_3$, which separate four qualitatively distinct phases and satisfy the following relationship for any $\sigma > 0$:

$$0 < \mu_1 = \frac{\sqrt{2\sigma^2 + 4\sigma + 4} - \sigma - 2}{\sigma^2} < \mu_2 = \frac{1}{\sigma + 4} < \mu_3 = \frac{\sigma + 1}{3\sigma + 4} < 1. \tag{4.4}$$

As $\mu$ passes any of these critical boundaries, one or more equilibria undergo a bifurcation, and the system's phase space is reconfigured. As demonstrated in appendix A.2, this behaviour is summarized by the following statements:

   I. If $\mu > \mu_3$, the game has one equilibrium, the uniform state $(1/2, 1/2)$. It is stable (a sink, i.e. stable in both dimensions).
   II. If $\mu_2 < \mu < \mu_3$, the game has three equilibria: the uniform state $(1/2, 1/2)$, which is unstable (a saddle, i.e. unstable in one dimension), and two internal equilibria satisfying $x_1 = \tilde{x}_2$ (and hence also $\tilde{x}_1 = x_2$). These are stable (sinks).
   III. If $\mu_1 < \mu < \mu_2$, the game has five equilibria: the above-mentioned three rest points as well as two further equilibria satisfying $x_1 = x_2$. The uniform state is unstable (a source, i.e. unstable in both dimensions), the equilibria satisfying $x_1 = \tilde{x}_2$ are stable (sinks) and the equilibria satisfying $x_1 = x_2$ are unstable (saddles).
   IV. If $\mu < \mu_1$, the game has nine equilibria: the above-mentioned five equilibria, as well as four further rest points. The uniform state is unstable (a source), the diagonal equilibria (i.e. those satisfying either $x_1 = \tilde{x}_2$ or $x_1 = x_2$) are stable (sinks), and the four new equilibria are unstable (saddles).

Figure 1 illustrates the bifurcation sequence.

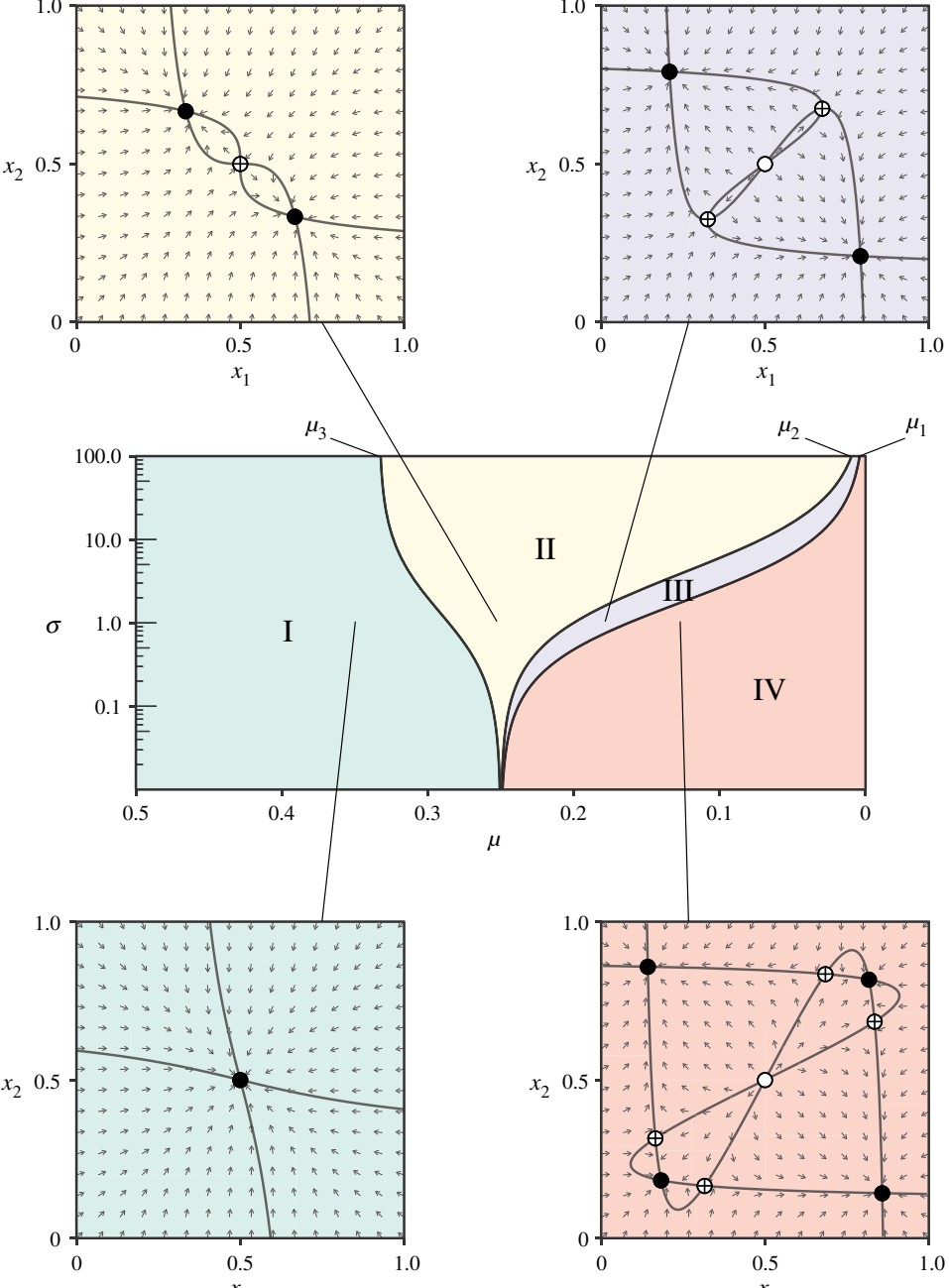

**Figure 1.** Bifurcation diagram of the fully symmetric two-strategy, two-population convergence–divergence game. The parameter space is divided into four phases, illustrated in the surrounding phase space plots for $\sigma = 1$ and the following values of $\mu$: 0.35, 0.25, 0.18, 0.13 (filled circles: sinks; crossed circles: saddles; open circles: sources). The curves indicate nullclines, i.e. the sets where either $\dot{x}_1 = 0$ or $\dot{x}_2 = 0$.

In particular, evolutionarily stable states of the fully symmetric convergence–divergence game with two strategies and two populations can assume one of three configurations. For high mutation rates (phase I), only the uniform state $(1/2, 1/2)$, in which each population uses each variant at equal frequencies, is stable. For slightly more moderate mutation rates (phase II), two non-uniform states are stable. Each of these satisfies $x_1 = \tilde{x}_2$ (and hence also $\tilde{x}_1 = x_2$), and could be termed a 'stable divergent state', since population 1 employs strategy 1 with the frequency with which population 2 employs strategy 2. In other words, in a stable divergent state, the two populations tend to adopt phenotypes that separate the populations to the greatest extent possible given the value of the mutation rate $\mu$ (higher mutation rates tending to lead to overall greater mixing). Phase III serves as a transition to phase IV, the phase of very low mutation rates, in which two more stable states appear. These satisfy $x_1 = x_2$ (and hence also $\tilde{x}_1 = \tilde{x}_2$), and could be

termed 'stable convergent states', as the two populations converge on the same frequencies with respect to both strategies. However, in phase IV the stable divergent states continue to be available, too.

To summarize, the qualitative behaviour of the two-strategy two-population convergence–divergence game can be characterized completely, if the effects of both social alignment and mutation are symmetric. The game has either one, two or four stable equilibria depending on the values of the social alignment and mutation rate parameters $\sigma$ and $\mu$. If real-life speech communities play such a game, and play it for long enough, they are then predicted to fall in one of these theoretically established stable rest points (modulo stochastic effects not explicitly modelled by the deterministic game). It is to this question of empirical adequacy that we next turn.

# 5. Jock and burnout identity dynamics

Convergence, divergence and identity dynamics are exhibited particularly clearly in adolescent sociolinguistic behaviour. Adolescence marks a sharp qualitative transition from childhood, where linguistic role models mostly come from the immediate family, to a new kind of social order dominated by peer-to-peer interactions [25]. A split typically occurs at this point whereby adolescents overwhelmingly identify with one of two social groups, often called 'jocks' and 'burnouts' in the North American context [51]. The social life of the prototypical jock is centred around school activities, while the prototypical burnout displays an aversion to school. These contrasting behaviours have been argued to represent different social adaptations to different needs, expectations and opportunities, considering that jocks are overwhelmingly destined for middle-class and burnouts for working-class trajectories [25]. Importantly, this social stratification coexists with a corresponding stratification in language use, with burnouts attesting innovative linguistic variants at higher frequencies than jocks. In other words, the two social groups exhibit socially motivated linguistic divergence.

In a detailed sociophonetic study, Eckert [25] explored the realization of the STRUT vowel /ʌ/ (found in words such as *cup*, *but* and *strut*) in jock and burnout groups in a Michigan high school in the 1980s. This vowel was undergoing backing (retraction in phonetic space) at the time as part of the wider sound change known as the Northern Cities Shift [58], with backed variants of the vowel considered innovative. As suggested above, burnouts were then expected to show higher rates of /ʌ/ backing than jocks, given their tendency to eschew conventional norms.[4] This is exactly what was found: Eckert [25, p. 201] reports the probability of backing of /ʌ/ in burnouts at 0.59, while in jocks the figure is only 0.43; a difference that was found statistically significant. This profile, $\mathbf{e} = (0.43, 0.59)$ (for Eckert), is satisfyingly close to the state $\mathbf{e}_m = (0.42, 0.58)$ (for 'Eckert modelled'), which by the results in §4 is a stable equilibrium of the fully symmetric convergence–divergence game for some combination of the social alignment and mutation rate parameters $\sigma$ and $\mu$. In the terminology introduced in §4, this is a stable divergent state.

One reason for choosing Eckert's [25] study as our empirical test case is that here it is possible to estimate the value of the mutation rate parameter $\mu$ using independent empirical evidence. Table 1 displays a subset of the phonetic confusion matrices for American listeners presented in [38]. Of interest is the total probability of confusing [ʌ] with any of [ɑ], [ɔ] and [ʊ], which are the vowels Eckert [25] considers backed variants of /ʌ/. From these data, we can compute the probability of the mutation ʌ → {ɑ, ɔ, ʊ} as follows, treating initial and final occurrences of the vowels as equally likely:

$$0.5 \cdot (0.125 + 0.083 + 0.012) + 0.5 \cdot (0.114 + 0.111 + 0.09) \approx 0.23. \tag{5.1}$$

In other words, experimental data on the confusability of vowels in American English by native listeners suggests setting $\mu = 0.23$ as a realistic value of the mutation rate parameter.

In §4, it was established that stable divergent states exist in each of the phases II–IV of the fully symmetric convergence–divergence game, but not in phase I. Furthermore, in phases II and III these kinds of states are the system's only stable equilibria. Revisiting figure 1, we find that the empirically estimated value of the mutation rate, $\mu = 0.23$, never cuts into phase I of the game, for any value of the social alignment parameter $\sigma$. Moreover, for sufficiently high values of $\sigma$, the system necessarily lands in either phase II or phase III given this value of $\mu$, and hence stable divergence is absolutely predicted.

Since the backing of /ʌ/ was part of an ongoing sound change, is it not misguided to model its empirical profile in jocks and burnouts as a stable equilibrium of the formal game? While this may seem genuinely problematic at first sight, closer examination of the timescales involved helps to

---

[4]Those conventional norms are, of course, largely set by adults. This point will be revisited in detail in §6.

**Table 1.** Confusion probabilities (as percentages) for four vowels of American English in native listeners, in initial (VC) and final (CV) positions, pooled across participants and across consonant contexts. From ([38], tables III–IV).

| stimulus | response (VC position) | | | | response (CV position) | | | |
|---|---|---|---|---|---|---|---|---|
| | ʌ | ɑ | ɔ | ʊ | ʌ | ɑ | ɔ | ʊ |
| ʌ | 64.9 | 12.5 | 8.3 | 1.2 | 65.3 | 11.4 | 11.1 | 0.9 |
| ɑ | 12.5 | 42.3 | 26.8 | 0.0 | 24.4 | 33.5 | 27.0 | 0.3 |
| ɔ | 4.5 | 36.3 | 47.3 | 1.2 | 3.7 | 23.9 | 65.3 | 0.6 |
| ʊ | 14.0 | 2.1 | 2.1 | 63.7 | 21.6 | 2.0 | 0.6 | 68.2 |

dissolve the worry. Even the most conservative estimates of the duration of the Northern Cities Shift measure it at the scale of speaker generations, and there is evidence that the earliest stages of the chain shift were already in operation in around 1900 [59]. Viewed against this backdrop of population-level diachronic change, processes of acquisition and use at the level of individuals are short-term, and the population itself appears to be, for all intents and purposes, at equilibrium.[5] In fact, it is natural to view the effects of the Northern Cities Shift on /ʌ/ not as a particular historical trajectory of the relevant speech community from some historical initial condition toward a point attractor, but rather as the movement of that point attractor itself in the system's phase space. This movement, in turn, can be construed as resulting from a change in the relevant mutation rate parameters, as I will argue in the immediately following section.

The results are thus encouraging: an empirically attested frequency profile corresponds to a stable equilibrium of the model for an independently plausible estimate of the relevant mutation rate parameter. Two residual problems remain, however. Firstly, the non-trivial equilibria of the fully symmetric game, i.e. equilibria other than the uniform state (1/2, 1/2), always appear in pairs. In phases II and III, for instance, not only one but two stable divergent states are available. The question then arises whether it is possible to explain to which one of these equilibria the empirical system converges, and why. Secondly, the validity of some of the strict symmetry assumptions employed above may be legitimately questioned. These problems will be addressed in the next section.

# 6. Adults and asymmetries

The above results illustrate, on an abstract level, an amount of agreement between a rather simple mathematical model and an empirically attested sociolinguistic pattern. The analysis so far is limited, however, in at least two respects. First, it is incontestable on empirical grounds that burnouts lead jocks in the backing of /ʌ/; not only is this what the data show, it is also expected given what is known about the social dynamics in this particular case [25]. As explained above, however, the model predicts two stable equilibria, one in which burnouts do lead, but also another in which the frequency profile is exactly reversed, with jocks showing the most backing. Furthermore, for sufficiently low mutation rates, stable states in which the two populations agree in their variant use are available. In an ideal situation, the model would predict not just the existence of these stable states, but also to which equilibrium the real-life system is expected to converge.

An empirically reasonable hypothesis is that burnouts' greater innovativeness may result from their greater desire to diverge from a third social group, namely, that of adults. If adults, in turn, should be conservative and attest only a limited degree of /ʌ/ backing, then the greater opposition that obtains between burnouts and adults (as opposed to the more amicable relationship between jocks and adults; see [25] for extended discussion) would lead us to expect that burnouts should, indeed, show a greater degree of innovativeness, and hence of /ʌ/ backing, than jocks. Although quantitative data on the frequency of /ʌ/ backing in the adult population is unfortunately not available, the assumption that adults represent the most conservative group is a reasonable position to hold in view of the general observation that adolescents tend to lead in the use of innovative linguistic forms [61].

---

[5]This position accrues further support from the empirical observation that people are rarely, if ever, aware of ongoing large-scale rotations of the vowel space, such as the Northern Cities Shift, even if they participate in them [60].

From a modelling point of view, the question then arises how to generalize the notion of extrinsic fitness—which above has been defined with two populations in mind—for multiple populations. Proceeding with the intuition that all social groups other than one's own ought to be treated as outgroups, and hence something to diverge from, we can generalize (3.7) for $N$ populations as follows:

$$\left. \begin{aligned} f_i(\mathbf{x}) &= s_{ii}x_i + \sum_{j=1, j \neq i}^{N} s_{ij}\tilde{x}_j x_i \\ \tilde{f}_i(\mathbf{x}) &= \tilde{s}_{ii}\tilde{x}_i + \sum_{j=1, j \neq i}^{N} \tilde{s}_{ij}x_j \tilde{x}_i \end{aligned} \right\} \quad (\mathbf{x} \in [0, 1]^N; \ i = 1, \ldots, N), \tag{6.1}$$

where the interpretation of the social alignment matrices $S = [s_{ij}]$ and $\tilde{S} = [\tilde{s}_{ij}]$ (now of size $N \times N$) remains as before. For $N = 3$ groups in particular, one has

and

$$\left. \begin{aligned} f_1(\mathbf{x}) &= s_{11}x_1 + s_{12}\tilde{x}_2 x_1 + s_{13}\tilde{x}_3 x_1, \\ f_2(\mathbf{x}) &= s_{22}x_2 + s_{21}\tilde{x}_1 x_2 + s_{23}\tilde{x}_3 x_2 \\ f_3(\mathbf{x}) &= s_{33}x_3 + s_{31}\tilde{x}_1 x_3 + s_{32}\tilde{x}_2 x_3 \end{aligned} \right\} \tag{6.2}$$

for the fitnesses $f_i$ (and mutatis mutandis for the complementary fitnesses $\tilde{f}_i$).

In the specific case of jocks, burnouts and adults, we also need to ask what shape the social alignment matrices should assume. With no evidence to the contrary, I will continue to assume $S = \tilde{S}$, i.e. that the strength of convergence/divergence does not depend on the identity of the linguistic variant. Writing $x_1$ for the frequency of /ʌ/ backing in jocks, $x_2$ for that in burnouts and $x_3$ for that in adults, a reasonable form for the matrix $S$ is then

$$S = \begin{pmatrix} 1 & \sigma & \tau_1 \\ \sigma & 1 & \tau_2 \\ \upsilon & \upsilon & 1 \end{pmatrix}, \tag{6.3}$$

where the diagonal is set to unity with the help of proposition A.2 (appendix A.2). To interpret this, note that $\sigma$ remains to denote the strength of divergence between jocks and burnouts (assumed symmetric for simplicity). The new $\tau_1$ and $\tau_2$ parameters supply the strength of divergence of the jock and burnout populations from the adult population, respectively, while $\upsilon$ gives the tendency of adults to diverge from the adolescent populations (assumed equal for both adolescent populations, given no evidence to the contrary). The empirically interesting scenario concerns the subset of the parameter space where $\tau_1 < \tau_2$.[6]

Returning now to the second problem identified above, we may ask whether it is reasonable to assume strict strategy symmetry to hold in this particular empirical situation. The mutation rate parameters in the replicator–mutator equation (3.5) were intended to model all kinds of sources of linguistic mutation not attributable to the (social) dynamics of convergence and divergence. But is it reasonable to assume that, setting those social motivations aside, the probability of backing /ʌ/ equals the probability of fronting its backed variants? As was mentioned in §5, the particular variable here tracked took part in a chain shift whose various subprocesses stand in a complex relation of dependencies, as is well-known [58]. In other words, there was a system-internal pressure for /ʌ/ to retract in phonetic space, and this fact ought to be reflected in our model as a higher rate for the mutation ʌ → {ɑ, ɔ, ʊ} compared to the corresponding back-mutation {ɑ, ɔ, ʊ}→ ʌ (moreover, this system-internal pressure ought to increase over time as the chain shift progresses). Recall that the empirical frequency profile was $\mathbf{e} = (0.43, 0.59)$, i.e. the probability of /ʌ/ backing in jocks was 0.43 and in burnouts 0.59. While this is close to the prediction derived from the fully symmetric model, $\mathbf{e}_m = (0.42, 0.58)$, the difference is precisely in the expected direction: the empirical populations back /ʌ/ more than the model populations.

Perhaps even more importantly, however, there turns out to be good reasons for thinking that the reverse mutation {ɑ, ɔ, ʊ} → ʌ is less likely than the mutation ʌ → {ɑ, ɔ, ʊ} even in the absence of system-internal pressures such as the contribution of interacting changes in a chain shift. From table 1,

---

[6]For simplicity, I gloss over the effects of potential asymmetries and differences in the interaction rates within and among the different populations, assuming interaction rates to be equal. The elements of $S$ then directly encode the strength of convergence/divergence. Although some differences in interaction rates are to be expected, the particular social setting in this case (high school) ensures that all groups interact to a considerable extent. Moreover, since quantitative estimates of the strength of convergence/divergence parameters are not available, it is always possible to adjust these parameters to account for differences in interaction rates in the elements of the aggregate matrix $S$. More discussion of these complications follows in §7.

**Table 2.** Interpretation of model parameters, together with the values used to obtain numerical solutions.

| parameter | meaning | value(s) |
|---|---|---|
| $k$ | skewing factor | $1/5, \ldots, 5$ |
| $\sigma$ | adolescents-to-adolescents social alignment | $1/5, \ldots, 5$ |
| $\tau_1$ | jocks-to-adults social alignment | $k^{-1}\sigma$ |
| $\tau_2$ | burnouts-to-adults social alignment | $k\sigma$ |
| $\upsilon$ | adults-to-adolescents social alignment | $1/5, 1, 5$ |
| $m$ | probability of backing mutation $\Lambda \rightarrow \{\alpha, \mathrm{ɔ}, \upsilon\}$ | $0.23$ |
| $\tilde{m}$ | probability of fronting mutation $\{\alpha, \mathrm{ɔ}, \upsilon\} \rightarrow \Lambda$ | $0.13$ |

we can compute the probability of the fronting mutation (again assuming initial and final occurrences of the vowels to be equally likely) to be only

$$0.5 \cdot \frac{0.125 + 0.045 + 0.140}{3} + 0.5 \cdot \frac{0.244 + 0.037 + 0.216}{3} \approx 0.13. \tag{6.4}$$

Although an overall decrease in the mutation rate is not predicted to take the system away from the phases in which the relevant stable equilibria are attested (cf. figure 1), it is not intuitively clear how an asymmetry like this will affect the system's overall phase portrait.

Extending the convergence–divergence game with an additional population and relaxing the strict strategy symmetry assumption leads to a dynamical system which is potentially better able to account for the empirical facts, but also one that is analytically intractable. A partial understanding of its behaviour can, however, be gained using numerical solution methods. To this end, I next assume that $\upsilon$, the adults-to-adolescents social alignment parameter (see again the matrix in (6.3)), can take one of the values $\upsilon = 1/5$, $\upsilon = 1$ and $\upsilon = 5$, and that the adolescents-to-adults social alignment parameters $\tau_1$ and $\tau_2$ satisfy $\tau_1 = k^{-1}\sigma$ and $\tau_2 = k\sigma$ for some $k > 0$. For $k > 1$, burnouts then tend to diverge more from the adults than jocks do, corresponding to the case $\tau_1 < \tau_2$ which above was identified as the empirically pertinent scenario. To explore systematically the dependence of the dynamics on the values of these parameters, I vary $\sigma$ and $k$ in equally spaced steps from $1/5$ to $5$. The mutation parameters are set at $m = 0.23$ for the backing mutation $\Lambda \rightarrow \{\alpha, \mathrm{ɔ}, \upsilon\}$ and at $\tilde{m} = 0.13$ for the fronting mutation $\{\alpha, \mathrm{ɔ}, \upsilon\} \rightarrow \Lambda$ in each of the three populations. Table 2 summarizes the interpretation of all model parameters, as well as the values used to obtain the numerical solutions.

For each parameter combination, the system was set in a random initial state $\mathbf{x}(0) = (x_1(0), x_2(0), x_3(0))$ where the frequencies of $/\Lambda/$ backing in the jock and burnout populations ($x_1(0)$ and $x_2(0)$, respectively) were drawn uniformly at random from the interval $[0, 1]$, while the corresponding frequency in the adult population, $x_3(0)$, was drawn uniformly from the restricted interval $[0, 1/4]$. This corresponds to the assumption that, in the historically attested initial condition, adults were conservative and attested lower frequencies of the innovation, while placing no restrictions on the potential starting points of the adolescent populations. The differential equations were then solved using the Dormand–Prince method [62] for 100 such randomly drawn initial states (for each individual parameter combination) until a stable equilibrium was reached.

The question of interest concerns what regions of the parameter space support not only the existence of a stable divergent state with the property $x_1 < 1/2 < x_2$ (burnouts attesting more backing of $/\Lambda/$ than jocks), but also the *necessary* convergence of the metapopulation into such a stable state. In other words: what proportion of the 100 solutions converged to a state satisfying $x_1 < 1/2 < x_2$ for any given combination of $\sigma$ and $k$? Figure 2 answers this question. Specifically, we find that if both $\sigma$ and $k$ have values roughly larger than 3, most or all instances of the system converge to the desired stable state. The adults-to-adolescents divergence parameter $\upsilon$ plays only a minor role, although it can be said that of the three values examined, $\upsilon = 1$ guarantees the widest convergence to the desired stable state. We can thus summarize these results as follows: if, and only if, jocks tend to diverge from adults less than they diverge from burnouts, and burnouts tend to diverge from adults more than they diverge from jocks, the system is guaranteed to converge to a stable divergent state in which burnouts attest more use of the innovative variant than jocks. This is fully in line with the empirical expectation.

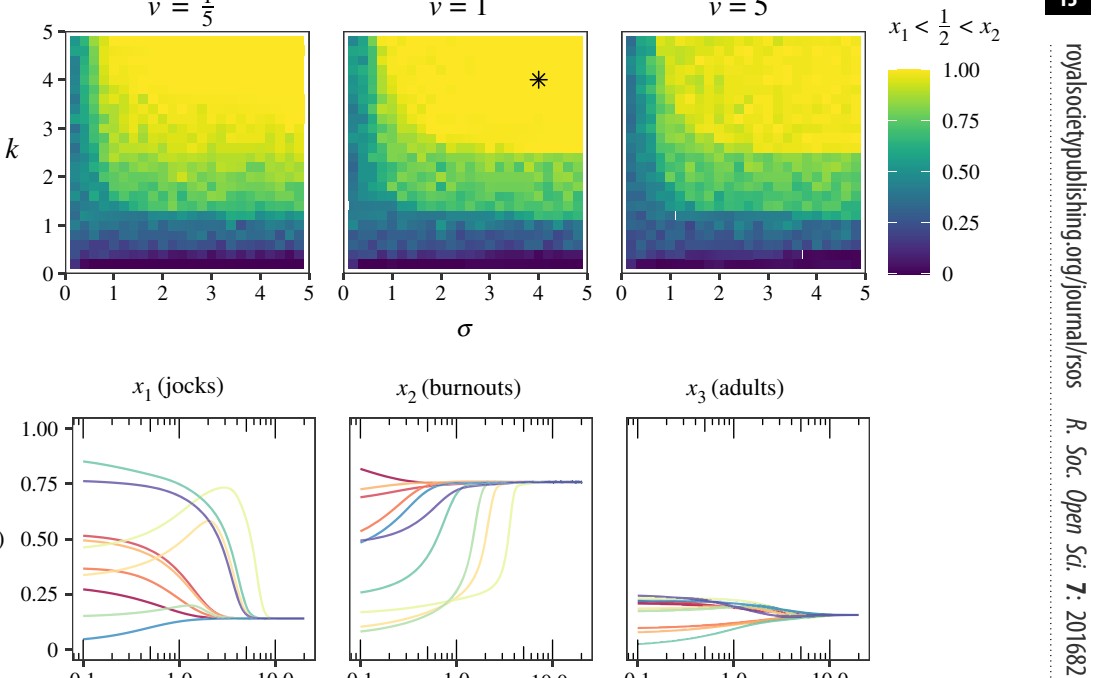

**Figure 2.** (a) Proportion of numerical solutions converging on a stable divergent state in which burnouts show more use of the innovative variant than jocks ($x_1 < 1/2 < x_2$), for various choices of the social alignment parameters (cf. table 2). (b) Ten randomly selected solutions from the ensemble of solutions with parameters indicated by the asterisk in (a).

## 7. Discussion

I have suggested that classical, macrosociolinguistic theory can usefully be complemented by a simple mathematical modelling paradigm in which structured but well-mixing populations interact and the fate of linguistic strategies depends, in part at least, on these social dynamics (extrinsic fitness). In particular, convergence to ingroup and divergence from outgroup linguistic behaviour can be modelled as an evolutionary game played by two or more populations, in which the fitness of a strategy is a nonlinear function of its usage frequencies in the populations on the one hand, and parameters controlling the strength of interaction and convergence/divergence within and between populations on the other (§§2 and 3). Section 4 demonstrated how the behaviour of this model can be fully analysed if strong symmetry assumptions are made. Application of the model to empirical data showed qualitative agreement (§5), in the sense that the empirically attested sociolinguistic pattern was found to correspond to a stable equilibrium of the model for an independently estimated value of the relevant mutation rate parameter. Section 6 additionally illustrated how the model's predictions can be sharpened through numerical exploration of an asymmetric multi-population extension of the game. In this concluding section, I discuss some of the remaining challenges, as well as prospects for future work along the above established lines.

Even in its simplest, symmetric two-population instantiation, application of the convergence–divergence game to empirical data requires the estimation of two parameters, a mutation rate parameter $\mu$ and a social alignment parameter $\sigma$. In the specific application here, it was possible to calibrate $\mu$, but data on $\sigma$ are lacking. It is important to be clear about what manner of corroboration may be expected in this sort of empirical application scenario, however. Given the very idealized character of the model, the expectation is not that it should predict the exact numbers of an empirical data point, but rather the general *form* of the data point, and hopefully to do this robustly over a range of possible model parameter values, so that the prediction does not depend on ad hoc values of the parameters. Thus, what from the point of view of modelling is intriguing about Eckert's [25] empirical frequency profile $\mathbf{e} = (0.43, 0.59)$ is not so much the fact that it comprises these particular frequencies, but that it is what above I have called a divergent state, i.e. it satisfies $e_1 < 1/2 < e_2$. The crucial question then is whether this nature of the empirical data point is predicted by the model (it is), as well as whether the prediction

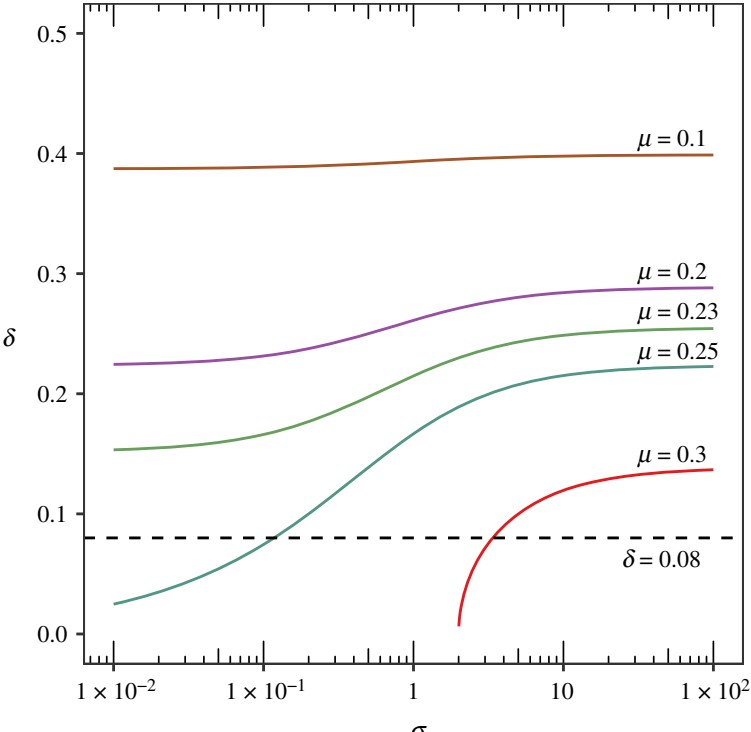

**Figure 3.** Displacement $\delta$ of the stable divergent states from the uniform state $(x_1, x_2) = (0.5, 0.5)$ in the fully symmetric game, for various values of mutation rate $\mu$ and social alignment $\sigma$. Each value of $\mu$ defines an envelope of possible locations for the equilibria. The dashed line gives the empirical displacement of the Eckert state $\mathbf{e}_m = (0.42, 0.58)$.

remains robust over a range of the model's parameter space (it does). It would in principle be possible to obtain estimates of a quantity such as $\sigma$, as an existing empirical literature on social biases demonstrates [63]. The practical challenges of conducting an experiment in which (i) the frequencies of use of a variable of interest in a number of populations, (ii) relevant parameters of the linguistic dynamics, such as the mutation rate, and (iii) social alignment parameters such as ingroup and outgroup biases can all be measured at the same time, are however considerable.

It is also possible to consider stricter, quantitative forms of empirical evaluation and to try to triangulate the value of the social alignment parameter $\sigma$. From equation (A 7) in appendix A.2, it is possible to deduce, in the fully symmetric special case, an envelope of values for $\sigma$ in which a given on-diagonal equilibrium $\mathbf{x}^* = (x_1^*, x_2^*)$ exists for any given pair of mutation rate $\mu$ and displacement $\delta = \delta_{\hat{D}}$, the latter defined as $\delta = |0.5 - x_1^*|$ (equivalently, $\delta = |0.5 - x_2^*|$). In the above case study of /ʌ/ backing, where $\mathbf{x}^* = \mathbf{e}_m = (0.42, 0.58)$ and consequently $\delta = 0.08$, it turns out that the modelled equilibrium in fact falls somewhat outside this envelope, assuming the above-estimated value $\mu = 0.23$ for the mutation rate parameter (figure 3). The empirical displacement does fall, however, within the envelope for $\mu = 0.25$. This predicts a value of $\sigma$ of the order of $10^{-1}$, implying that convergence to ingroup norm is roughly an order of magnitude stronger than divergence from outgroup norm in the specific case of jocks and burnouts (again assuming uniform interaction rates within and between the two groups). This accords well, in principle, with the results of empirical studies of intra- and inter-group cooperation which have found ingroup favouritism to result from a pro-ingroup rather than an anti-outgroup bias [63]. It is important to bear in mind, however, the uncertainty inherent in empirical data points such as the frequency profile $\mathbf{e}_m$ or the estimate of the mutation rate $\mu$. For one, Eckert's [25] data include a considerable amount of inter-individual variability within each population; this, together with the small size of the populations, implies that some amount of noise in $\mathbf{e}_m$ (and, consequently, in the displacement $\delta$) is to be expected. Similarly, the phoneme confusion probabilities reported in [38] are averages over participants and presentations of vowel tokens, and so involve some unknown amount of uncertainty. In connection with further empirical testing, it would also be important to consider other sociolinguistic variables (such as those reported in [64] for jocks and burnouts), but also other speech communities so as not to overfit the model to the specifics of one particular sociolinguistic situation.

Apart from the above empirical considerations, it would be possible to explore various formal extensions and refinements of the modelling approach adopted in this paper. One of these concerns the resolution at which populations are approached—for technical reasons, I have here assumed infinite, homogeneous well-mixing populations. As noted above, Eckert's [25] data, in fact, show an amount of inter-individual variability within each social group; the assumption of homogeneous populations makes modelling this aspect of the data impossible. Established techniques exist, however, for studying evolutionary games in finite and inhomogeneous populations [65]; within-population variability may also be modelled using stochastic dynamics instead of the deterministic setting here assumed [66]. Conceptually straightforward ways also exist of generalizing the convergence–divergence game for multiple strategies, so that modelling the dynamics of more than two linguistic variants in a number of populations becomes possible. Other promising avenues for future work include analytical treatments of near-symmetric variations of the game, i.e. models in which one aspect of the symmetry (such as mutation symmetry) is relaxed but others kept in place, and an exploration of possible ways of interpreting the population-level results here reported from the vantage point of individual-level models, perhaps along the lines of [67].

In this paper, I have suggested that the identity dynamics of linguistic strategies in stratified but interacting populations of speakers can be illuminated by a formal model of linguistic convergence and divergence, couched in evolutionary game theory. In reality, processes of convergence and divergence attest a number of nuanced characteristics which the simple model here studied is unable to account for. Where the simple model assumes that an aggregate population-level frequency (such as the frequency of backing the vowel /ʌ/) is a useful proxy of individual-level linguistic behaviour, the real-life situation is far more complex than this. Notably, linguistic accommodation operates on multiple (time)scales and is responsible not just for large-scale regularities such as the population-level opposition observed between jock and burnout language use, but also for the ways in which people temporarily adjust and attune their linguistic choices, as demonstrated by the phenomenon of style shifting [13]. The model here proposed could naturally be extended to cater for these complexities for instance via the inclusion of socially motivated production biases [68]. Although analytical tractability is almost certainly lost, pursuing such extensions in future work is likely to lead to additional insights into the complex relationship that obtains between the dynamics of linguistic variants and the social dynamics of populations.

Data accessibility. This article has no additional data. Numerical solutions of the asymmetric three-population game in §6 were obtained using the Dormand–Prince method [62] implemented in Julia [69], v. 1.4.2. The Jacobian of the fully symmetric game, and its eigenvalues, were symbolically calculated using the Maxima Computer Algebra System. All code is available at https://doi.org/10.5281/zenodo.4034829.

Competing interests. I declare I have no competing interests.

Funding. The research here reported was funded by the Federal Ministry of Education and Research (BMBF) and the Baden-Württemberg Ministry of Science as part of the Excellence Strategy of the German Federal and State Governments.

Acknowledgements. I am grateful to Ricardo Bermúdez-Otero, George Walkden and the referees for comments.

# Appendix A

## A.1. The replicator–mutator equation

The $n$-strategy one-population continuous-time replicator–mutator equation [52] reads[7]

$$\dot{x}_i = \sum_{j=1}^{n} x_j m_{ji} f_j(\mathbf{x}) - x_i \phi(\mathbf{x}) \tag{A 1}$$

for the $i$th strategy, where $M = [m_{ij}]$ is a row-stochastic $n \times n$ matrix supplying the mutation rates from one strategy to another, and $\phi(\mathbf{x})$ is average fitness, defined as

$$\phi(\mathbf{x}) = \sum_{i=1}^{n} x_i f_i(\mathbf{x}). \tag{A 2}$$

---

[7]Note that here, in contrast to the rest of this paper, the subscripts index strategies instead of populations.

For two strategies ($n = 2$), the mutation matrix has only two free parameters:

$$M = \begin{pmatrix} m_{11} & m_{12} \\ m_{21} & m_{22} \end{pmatrix} = \begin{pmatrix} 1 - m_{12} & m_{12} \\ m_{21} & 1 - m_{21} \end{pmatrix}. \tag{A 3}$$

Writing $x = x_1$, $\tilde{x} = x_2$, $f = f_1$ and $\tilde{f} = f_2$ as above, and noting that the average fitness equals $\phi(\mathbf{x}) = xf(\mathbf{x}) + \tilde{x}\tilde{f}(\mathbf{x})$, equation (A 1) becomes

$$\begin{aligned}
\dot{x} &= xm_{11}f(\mathbf{x}) + \tilde{x}m_{21}\tilde{f}(\mathbf{x}) - x[xf(\mathbf{x}) + \tilde{x}\tilde{f}(\mathbf{x})] \\
&= (1 - m_{12} - x)xf(\mathbf{x}) + (m_{21} - x)\tilde{x}\tilde{f}(\mathbf{x}) \\
&= (1 - x - m_{12})xf(\mathbf{x}) - (x - m_{21})\tilde{x}\tilde{f}(\mathbf{x}) \\
&= (\tilde{x} - m_{12})xf(\mathbf{x}) - (x - m_{21})\tilde{x}\tilde{f}(\mathbf{x}).
\end{aligned} \tag{A 4}$$

Writing $\tilde{m}$ for $m_{12}$ and $m$ for $m_{21}$ recovers the form used in the main body of the paper.

## A.2. Proofs

This appendix provides proofs of propositions 3.1 and 4.1. Proposition 3.1 is reproduced below as proposition A.1; proposition 4.1 follows from the more general result stated as proposition A.2. Finally, the bifurcation sequence of the fully symmetric game is stated and proved as proposition A.5.

**Proposition A.1.** *The number of equilibria of the two-strategy, two-population convergence–divergence game is at least one and at most nine.*

*Proof.* The existence of at least one rest point follows from Brouwer's fixed point theorem.

For the upper bound, plugging the fitnesses (3.7) into (3.5) shows that $\dot{x}_1$ and $\dot{x}_2$ are cubic in $x_1$ and $x_2$. In particular, each nullcline is a cubic curve in the $x_1x_2$ plane. By Bézout's theorem, the number of nullcline intersections, and hence of equilibria, is then at most nine. ∎

**Proposition A.2.** *Let $S = \tilde{S}$ be an $N \times N$ social alignment matrix without zeroes. Then the elements of S can always be rescaled so that the diagonal of S is 1, without affecting the orbits of the resulting dynamical system under fitnesses (6.1) and the dynamic (3.5).*

*Proof.* Assume fitnesses $f_i$ and $\tilde{f}_i$ with $S = \tilde{S}$, and let $F_i(\mathbf{x}) = \alpha f_i(\mathbf{x})$ and $\tilde{F}_i(\mathbf{x}) = \alpha \tilde{f}_i(\mathbf{x})$ where $\alpha = 1/s_{ii}$. Consider the differential equations

$$\dot{x}_i = h_i(\mathbf{x}) \quad \text{and} \quad \dot{x}_i = H_i(\mathbf{x}),$$

where

$$h_i(\mathbf{x}) = (\tilde{x}_i - \tilde{m}_i)x_if_i(\mathbf{x}) - (x_i - m_i)\tilde{x}_i\tilde{f}_i(\mathbf{x})$$

and

$$H_i(\mathbf{x}) = (\tilde{x}_i - \tilde{m}_i)x_iF_i(\mathbf{x}) - (x_i - m_i)\tilde{x}_i\tilde{F}_i(\mathbf{x}).$$

Then

$$H_i(\mathbf{x}) = \alpha[(\tilde{x}_i - \tilde{m}_i)x_if_i(\mathbf{x}) - (x_i - m_i)\tilde{x}_i\tilde{f}_i(\mathbf{x})] = \alpha h_i(\mathbf{x}),$$

i.e. use of $H_i$ instead of $h_i$ amounts to nothing but a rescaling of time by a factor of $\alpha$. ∎

To prove proposition A.5 on the bifurcation sequence of the fully symmetric game, I make use of the following two lemmas.

**Lemma A.3.** *Under full symmetry, the uniform state $(x_1, x_2) = (1/2, 1/2)$ is an equilibrium of the two-strategy, two-population convergence–divergence game for any combination of $\mu$ and $\sigma$.*

*Proof.* First, note that $\tilde{x}_1 = x_1$ and $\tilde{x}_2 = x_2$ if $(x_1, x_2) = (1/2, 1/2)$. Hence $f_i(\mathbf{x}) = \tilde{f}_i(\mathbf{x})$ from (4.2), and consequently $\dot{x}_i = (\tilde{x}_i - \mu)x_if_i(\mathbf{x}) - (x_i - \mu)\tilde{x}_i\tilde{f}_i(\mathbf{x}) = 0$. ∎

**Lemma A.4.** *Assume full symmetry. Then the following are equivalent in the two-strategy, two-population convergence–divergence game:*

(i) *$(x_1, x_2)$ is an equilibrium.*
(ii) *$(x_2, x_1)$ is an equilibrium.*

(iii) $(\tilde{x}_2, \tilde{x}_1)$ is an equilibrium.

(iv) $(\tilde{x}_1, \tilde{x}_2)$ is an equilibrium.

*Proof.* That $(x_1, x_2)$ is an equilibrium if and only if $(x_2, x_1)$ is an equilibrium follows immediately with the change of variables $x_1 \leftrightarrow x_2$. It then suffices to show that $(x_1, x_2)$ is an equilibrium if and only if $(\tilde{x}_1, \tilde{x}_2) = (1 - x_1, 1 - x_2)$ is an equilibrium. For this, notice that $\tilde{\tilde{x}}_i = 1 - (1 - x_i) = x_i$ and that $\tilde{f}_i(\mathbf{x}) = f_i(\tilde{\mathbf{x}})$ and $f_i(\mathbf{x}) = \tilde{f}_i(\tilde{\mathbf{x}})$, where $\tilde{\mathbf{x}} = (\tilde{x}_1, \tilde{x}_2)$. Thus $\dot{x}_i = 0$ if and only if $(\tilde{x}_i - \mu)x_i f_i(\mathbf{x}) = (x_i - \mu)\tilde{x}_i \tilde{f}_i(\mathbf{x})$ if and only if $(\tilde{x}_i - \mu)\tilde{\tilde{x}}_i \tilde{f}_i(\tilde{\mathbf{x}}) = (\tilde{\tilde{x}}_i - \mu)\tilde{x}_i f_i(\tilde{\mathbf{x}})$ if and only if $\dot{\tilde{x}}_i = 0$. Hence $(x_1, x_2)$ is an equilibrium if and only if $(\tilde{x}_1, \tilde{x}_2)$ is. ∎

We can now state and prove proposition A.5.

**Proposition A.5.** *A two-strategy, two-population convergence–divergence game with full symmetry has either 1, 3, 5 or 9 equilibria depending on the value of the mutation rate and social alignment parameters $\mu$ and $\sigma$. These phases are separated by three critical values of $\mu$—$\mu_1, \mu_2$ and $\mu_3$—satisfying the following relationship for any $\sigma > 0$:*

$$0 < \mu_1 = \frac{\sqrt{2\sigma^2 + 4\sigma + 4} - \sigma - 2}{\sigma^2} < \mu_2 = \frac{1}{\sigma + 4} < \mu_3 = \frac{\sigma + 1}{3\sigma + 4} < 1. \tag{A 5}$$

*At each of these critical boundaries, one or two equilibria of the system are non-hyperbolic and undergo a bifurcation, giving rise to the following four phases:*

I. *For $\mu > \mu_3$, the system has one equilibrium, the uniform state $(1/2, 1/2)$, which is stable (a sink). In the transition from phase I to phase II, the uniform state undergoes a bifurcation, giving rise to two further equilibria:*

II. *For $\mu_2 < \mu < \mu_3$, the system has three equilibria, the uniform state $(1/2, 1/2)$, which is unstable (a saddle), and two equilibria on the diagonal*

$$\tilde{D} := \{(x_1, x_2) \in [0, 1]^2 : x_1 = \tilde{x}_2\}, \tag{A 6}$$

*which are both stable (sinks). The on-diagonal equilibria are $(x_1, x_2) = (1/2 + \delta_{\tilde{D}}, 1/2 - \delta_{\tilde{D}})$ and $(x_1, x_2) = (1/2 - \delta_{\tilde{D}}, 1/2 + \delta_{\tilde{D}})$ with the displacement from $1/2$ given by*

$$\delta_{\tilde{D}} = \frac{\sqrt{-(3\sigma^2 + 4\sigma)\mu^2 - 2(\sigma^2 + 3\sigma + 2)\mu + (\sigma + 1)^2}}{2(\sigma\mu + \sigma + 1)}. \tag{A 7}$$

*In the transition from phase II to phase III, the uniform state $(1/2, 1/2)$ undergoes a second bifurcation, giving rise to two equilibria on the other diagonal:*

III. *For $\mu_1 < \mu < \mu_2$, the system has five equilibria, the above-mentioned uniform state and $\tilde{D}$-equilibria, as well as two further equilibria on the diagonal*

$$D := \{(x_1, x_2) \in [0, 1]^2 : x_1 = x_2\}, \tag{A 8}$$

*namely $(x_1, x_2) = (1/2 + \delta_D, 1/2 + \delta_D)$ and $(x_1, x_2) = (1/2 - \delta_D, 1/2 - \delta_D)$ with displacement*

$$\delta_D = \frac{\sqrt{(\sigma^2 + 4\sigma)\mu^2 - 2(\sigma + 2)\mu + 1}}{2(\sigma\mu - 1)}. \tag{A 9}$$

*The uniform state is unstable (now a source), the $\tilde{D}$-equilibria are stable (sinks) and the D-equilibria are unstable (saddles). In the transition from phase III to phase IV, both D-equilibria undergo a bifurcation, giving rise to four further rest points:*

IV. *For $\mu < \mu_1$, the system has nine equilibria, the above-mentioned uniform state, $\tilde{D}$-equilibria and D-equilibria, as well as four off-diagonal equilibria whose values are too complicated to calculate (except in the border case $\mu = 0$). The uniform state is unstable (a source), the $\tilde{D}$-equilibria are stable (sinks), the D-equilibria are stable (sinks) and the off-diagonal equilibria are unstable (saddles).*

*Proof.* For notational convenience, I shall write $x = x_1$, $y = x_2$, $f = f_1$ and $g = f_2$ and omit the arguments on the fitnesses in what follows.

To make a start, let us consider the zero-mutation special case $\mu = 0$. The system (4.3) then reduces to the pair of replicator equations

and

$$\left.\begin{array}{l} \dot{x} = x\tilde{x}(f - \tilde{f}) \\ \dot{y} = y\tilde{y}(g - \tilde{g}). \end{array}\right\}$$

From this, it is obvious that each vertex of the unit square $[0, 1]^2$ is an equilibrium in addition to the uniform equilibrium $(1/2, 1/2)$ guaranteed by lemma A.3. Further equilibria exist if $f = \tilde{f}$ or $g = \tilde{g}$ or both. From (4.2), solving $f = \tilde{f}$ for $x$ yields

$$x = \frac{1 + \sigma y}{2 + \sigma}.$$

In particular,

$$x|_{y=0} = \frac{1}{2 + \sigma} =: \xi_0 \quad \text{and} \quad x|_{y=1} = \frac{1 + \sigma}{2 + \sigma} =: \xi_1,$$

so $(\xi_0, 0)$ and $(\xi_1, 1)$ are rest points. Symmetrically,

$$y|_{x=0} = \frac{1}{2 + \sigma} =: \upsilon_0 \quad \text{and} \quad y|_{x=1} = \frac{1 + \sigma}{2 + \sigma} =: \upsilon_1$$

are solutions of $g = \tilde{g}$. Hence $(0, \upsilon_0)$ and $(1, \upsilon_1)$ are also equilibria. By proposition 3.1, there are no further equilibria.

To find out the stabilities of these rest points, we study the system's linearization around each of them. For general $\mu$, the Jacobian of (4.3) with fitnesses (4.2) is[8]

$$J = \begin{pmatrix} f[2(\tilde{x} - \mu) - x] + \tilde{f}[2(x - \mu) - \tilde{x}] & -\sigma[\tilde{x}^2(x - \mu) + x^2(\tilde{x} - \mu)] \\ -\sigma[\tilde{y}^2(y - \mu) + y^2(\tilde{y} - \mu)] & g[2(\tilde{y} - \mu) - y] + \tilde{g}[2(y - \mu) - \tilde{y}] \end{pmatrix}.$$

With $\mu = 0$, its eigenvalues at the above nine rest points are as follows:

(i) At $(0, 0)$ and $(1, 1)$: $-1$ (repeated)
(ii) At $(0, 1)$ and $(1, 0)$: $-(1 + \sigma)$ (repeated)
(iii) At $(1/2, 1/2)$: $1/2$ and $(1 + \sigma)/2$
(iv) At $(\xi_0, 0)$, $(\xi_1, 1)$, $(0, \upsilon_0)$ and $(1, \upsilon_1)$: $(1 + \sigma)/(2 + \sigma)$ and $-2(1 + \sigma)/(2 + \sigma)$.

Bearing in mind that $\sigma > 0$, we thus find that the vertex rest points are all sinks, that the uniform state $(1/2, 1/2)$ is a source, and that the remaining four rest points on the edges of the unit cube are saddles.

As the value of the mutation rate parameter $\mu$ is increased, the expectation is that these equilibria will gradually vanish, so that at the limit of maximal mutation only the uniform equilibrium remains (cf. similar treatments of replicator–mutator dynamics with linear fitness in [26,27,55,56]). To study this bifurcation scenario in detail, I next assume that, under full symmetry, it is reasonable to expect at least some of these equilibria to fall on the diagonals of the unit square,

$$D = \{(x, y) \in [0, 1]^2 : x = y\} \quad \text{and} \quad \tilde{D} = \{(x, y) \in [0, 1]^2 : x = \tilde{y}\}.$$

Note that the existence of an on-diagonal equilibrium $(x, y) \in (D \cup \tilde{D}) \setminus \{(1/2, 1/2)\}$ which is not the uniform state $(1/2, 1/2)$ implies the existence of one further equilibrium (namely, its reflection about the other diagonal), and that the existence of an off-diagonal equilibrium $(x, y) \in [0, 1]^2 \setminus (D \cup \tilde{D})$ implies the existence of three further equilibria (its reflections about each diagonal) by lemma A.4. Since the number of equilibria is between 1 and 9 by proposition 3.1, it follows that the number of equilibria for any combination of $\mu$ and $\sigma$ has to be either 1, 3, 5, 7 or 9. This suggests a sequence of pitchfork bifurcations as $\mu$ decreases from 1 towards 0.

On-diagonal equilibria are not difficult to find algebraically. Upon the relevant substitutions in (4.3), direct mechanical application of the cubic formula yields non-trivial roots $x_{\pm D} = 1/2 \pm \delta_D$ for $\dot{x}|_{y=x}$ with

$$\delta_D = \frac{\sqrt{(\sigma^2 + 4\sigma)\mu^2 - 2(\sigma + 2)\mu + 1}}{2(\sigma\mu - 1)} = \frac{\sqrt{A(\sigma, \mu)}}{2(\sigma\mu - 1)},$$

where I write $A(\sigma, \mu)$ for the argument of the square root. Due to symmetry, $\dot{y}|_{x=y}$ has the same roots. Hence there are a maximum of two non-trivial ($\delta_D \neq 0$) rest points on the $D$-diagonal, and these are $\mathbf{d}_1 := (x_{+D}, y_{+D})$ and $\mathbf{d}_2 := (x_{-D}, y_{-D})$ provided that the roots exist in the reals and in particular in the

---

[8]I omit details of purely mechanical calculation, such as the computation of the Jacobian and its eigenvalues. A symbolic implementation of these calculations is available (see 'Data accessibility').

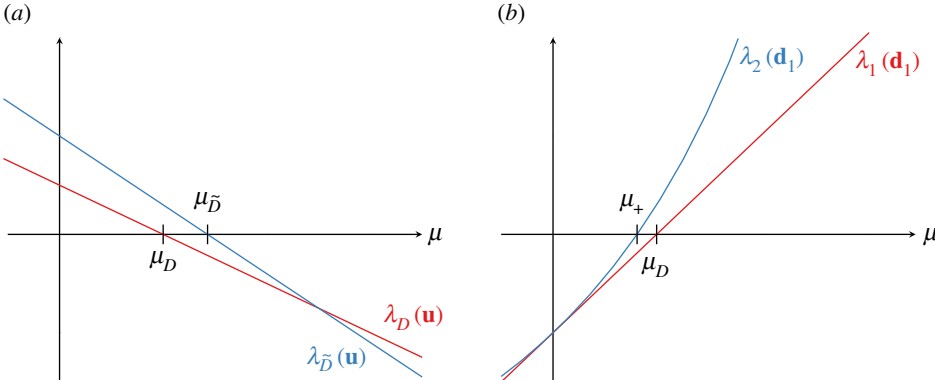

**Figure 4.** (*a*) The eigenvalues $\lambda_D(\mathbf{u})$ and $\lambda_{\tilde{D}}(\mathbf{u})$ of the Jacobian, evaluated at the uniform rest point $\mathbf{u} = (1/2, 1/2)$, as a function of mutation rate $\mu$. (*b*) The eigenvalues $\lambda_1(\mathbf{d}_1)$ and $\lambda_2(\mathbf{d}_1)$ of the Jacobian, evaluated at the diagonal rest point $\mathbf{d}_1$ (see text), as a function of mutation rate $\mu$.

interval $[0, 1]$. On the other diagonal, the solutions are $x_{\pm\tilde{D}} = y_{\pm\tilde{D}} = 1/2 \pm \delta_{\tilde{D}}$ with

$$\delta_{\tilde{D}} = \frac{\sqrt{-(3\sigma^2 + 4\sigma)\mu^2 - 2(\sigma^2 + 3\sigma + 2)\mu + (\sigma + 1)^2}}{2(\sigma\mu + \sigma + 1)} = \frac{\sqrt{\tilde{A}(\sigma, \mu)}}{2(\sigma\mu + \sigma + 1)}.$$

It follows that the $\tilde{D}$-diagonal equilibria are $\tilde{\mathbf{d}}_1 := (x_{+\tilde{D}}, y_{-\tilde{D}})$ and $\tilde{\mathbf{d}}_2 := (x_{-\tilde{D}}, y_{+\tilde{D}})$.

The non-trivial $D$-equilibria exist whenever $A(\sigma, \mu) > 0$ and $\delta_D$ satisfies $|\delta_D| \leq 1/2$. It is easy to verify that $|\delta_D| \leq 1/2$ if and only if $\mu < 1/\sigma$. On the other hand, $A(\sigma, \mu)$ is an upward-opening parabola in $\mu$, and has roots $\mu_1 = 1/(\sigma + 4) > 0$ and $\mu_2 = 1/\sigma > \mu_1$ as can easily be verified. Thus, the $D$-equilibria $\mathbf{d}_1$ and $\mathbf{d}_2$ exist whenever $\mu < 1/(\sigma + 4)$. A similar argument concerning $\delta_{\tilde{D}}$ and $\tilde{A}(\sigma, \mu)$ shows that the $\tilde{D}$-equilibria $\tilde{\mathbf{d}}_1$ and $\tilde{\mathbf{d}}_2$ exist whenever $\mu < (\sigma + 1)/(3\sigma + 4)$. We have thus identified two critical values of the mutation parameter $\mu$,

$$\mu_D := \frac{1}{\sigma + 4} \quad \text{and} \quad \mu_{\tilde{D}} := \frac{\sigma + 1}{3\sigma + 4},$$

supplying upper bounds for the existence of non-trivial on-diagonal equilibria. Since

$$\frac{1}{\sigma + 4} < \frac{\sigma + 1}{3\sigma + 4}$$

for any $\sigma > 0$, the existence of the $D$-equilibria implies the existence of the $\tilde{D}$-equilibria.

The pairs of on-diagonal equilibria ($\mathbf{d}_1$ and $\mathbf{d}_2$, $\tilde{\mathbf{d}}_1$ and $\tilde{\mathbf{d}}_2$) bifurcate from the uniform equilibrium $\mathbf{u} = (1/2, 1/2)$. To see this, consider the eigenvalues of the Jacobian evaluated at the uniform equilibrium $\mathbf{u}$. In the fully symmetric case, these are easy enough to solve and are found to be

$$\lambda_D(\mathbf{u}) := -\frac{\sigma + 4}{2}\mu + \frac{1}{2} \quad \text{and} \quad \lambda_{\tilde{D}}(\mathbf{u}) := -\frac{3\sigma + 4}{2}\mu + \frac{1 + \sigma}{2}.$$

For fixed $\sigma$, each eigenvalue is thus affine in $\mu$, with roots

$$\frac{1}{\sigma + 4} = \mu_D \quad \text{and} \quad \frac{\sigma + 1}{3\sigma + 4} = \mu_{\tilde{D}},$$

respectively (figure 4*a*). For $\mu < \mu_D$, both eigenvalues are positive and $\mathbf{u}$ is a source. For $\mu_D < \mu < \mu_{\tilde{D}}$, $\lambda_{\tilde{D}}(\mathbf{u})$ is positive and $\lambda_D(\mathbf{u})$ negative, and $\mathbf{u}$ is a saddle. For $\mu_{\tilde{D}} < \mu$, both eigenvalues are negative, and $\mathbf{u}$ is a sink.

The existence of the (unstable) side-of-the-square equilibria $(\xi_0, 0)$, $(\xi_1, 1)$, $(0, \upsilon_0)$ and $(1, \upsilon_1)$ in the zero-mutation $\mu = 0$ case suggests that off-diagonal equilibria should exist in the $\mu > 0$ case too, possibly bifurcating from the equilibria on the $D$-diagonal as these approach the vertices as $\mu$ tends to zero. This is, indeed, the case. While it has not been possible to find an explicit solution of these equilibria, the bifurcation points can be identified relatively easily by evaluating the eigenvalues of the Jacobian

at the $D$-equilibria $\mathbf{d}_1$ and $\mathbf{d}_2$. Due to symmetry, it suffices to consider $\mathbf{d}_1$ only. The eigenvalues are found to be

$$\lambda_1(\mathbf{d}_1) = (\sigma + 4)\mu - 1 \quad \text{and} \quad \lambda_2(\mathbf{d}_1) = \frac{\sigma^2\mu^2 + 2(\sigma + 2)\mu - 1}{1 - \sigma\mu}.$$

Thus, $\lambda_1(\mathbf{d}_1)$ is affine in $\mu$ with a root at $\mu = 1/(\sigma + 4) = \mu_D$. As the $D$-equilibria only exist when $\mu < \mu_D$, it thus suffices to consider values of $\mu$ less than $1/(\sigma + 4)$, which also takes care of the singularity $\mu = 1/\sigma$ for $\lambda_2(\mathbf{d}_1)$. On the other hand, $\lambda_2(\mathbf{d}_1)$ has roots at

$$\mu_- := \frac{-(\sigma + 2) - \sqrt{2\sigma^2 + 4\sigma + 4}}{\sigma^2} \quad \text{and} \quad \mu_+ := \frac{-(\sigma + 2) + \sqrt{2\sigma^2 + 4\sigma + 4}}{\sigma^2}.$$

Of these, $\mu_-$ is always negative while $\mu_+$ is always positive with $\mu_+ \leq \mu_D$. The $D$-diagonal equilibrium undergoes a subcritical pitchfork bifurcation at $\mu_+$ as the second eigenvalue $\lambda_2(\mathbf{d}_1)$ changes sign as the value of $\mu$ is decreased (figure 4$b$). Consequently, this $\mu_+$ is the third critical value of the mutation parameter. Writing $\mu_1 = \mu_+$, $\mu_2 = \mu_D$ and $\mu_3 = \mu_{\tilde{D}}$, we are done. ∎

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
