## [Reviewer comments · Royal Society Open Science]

Review History

RSOS-201682.R0 (Original submission)

Review form: Reviewer 1

Is the manuscript scientifically sound in its present form?

Yes

Are the interpretations and conclusions justified by the results?

Yes

Is the language acceptable?

Yes

Do you have any ethical concerns with this paper?

No

Have you any concerns about statistical analyses in this paper?

No

Recommendation?

Accept as is

Comments to the Author(s)

This is an excellent paper. It draws on existing literature of evolutionary game-theoretic modeling of language dynamics, but it breaks considerable new ground. This applies both to the technical aspects (non-linear fitness function; explicit modeling of mutations) and the empirical coverage. The fact that the equilibrium analysis is linked with results from an empirical study from the literature is especially impressive.

Review form: Reviewer 2

Is the manuscript scientifically sound in its present form?

Yes

Are the interpretations and conclusions justified by the results?

Yes

Is the language acceptable?

Yes

Do you have any ethical concerns with this paper?

No

Have you any concerns about statistical analyses in this paper?

No

Recommendation?

Accept as is

Comments to the Author(s)

I found one tiny mistake: At the bottom of p21 line 58, I think you mean non-hyperbolic

Review form: Reviewer 3

Is the manuscript scientifically sound in its present form?

Yes

Are the interpretations and conclusions justified by the results?

Yes

Is the language acceptable?

Yes

Do you have any ethical concerns with this paper?

No

Have you any concerns about statistical analyses in this paper?

No

Recommendation?

Accept with minor revision (please list in comments)

Comments to the Author(s)

See attached file (Appendix A).

Decision letter (RSOS-201682.R0)

Dear Dr Kauhanen

On behalf of the Editors, we are pleased to inform you that your Manuscript RSOS-201682 "Replicator-mutator dynamics of linguistic convergence and divergence" has been accepted for publication in Royal Society Open Science subject to minor revision in accordance with the referees' reports. Please find the referees' comments along with any feedback from the Editors below my signature.

Please submit your revised manuscript and required files (see below) no later than 7 days from today's (ie 14-Oct-2020) date. Note: the ScholarOne system will 'lock' if submission of the revision is attempted 7 or more days after the deadline. If you do not think you will be able to meet this deadline please contact the editorial office immediately.

on behalf of Professor Matjaz Perc (Associate Editor) and Mark Chaplain (Subject Editor)
openscience@royalsociety.org

Reviewer comments to Author:

Reviewer: 1

Comments to the Author(s)

This is an excellent paper. It draws on existing literature of evolutionary game-theoretic modeling of language dynamics, but it breaks considerable new ground. This applies both to the technical

aspects (non-linear fitness function; explicit modeling of mutations) and the empirical coverage. The fact that the equilibrium analysis is linked with results from an empirical study from the literature is especially impressive.

Reviewer: 2

Comments to the Author(s)

I found one tiny mistake: At the bottom of p21 line 58, I think you mean non-hyperbolic

Reviewer: 3

Comments to the Author(s)

See attached file

===PREPARING YOUR MANUSCRIPT===

- one version identifying all the changes that have been made (for instance, in coloured highlight, in bold text, or tracked changes);
- a 'clean' version of the new manuscript that incorporates the changes made, but does not highlight them. This version will be used for typesetting.

===PREPARING YOUR REVISION IN SCHOLARONE===

Author's Response to Decision Letter for (RSOS-201682.R0)

See Appendix B.

Decision letter (RSOS-201682.R1)

Dear Dr Kauhanen,

It is a pleasure to accept your manuscript entitled "Replicator–mutator dynamics of linguistic convergence and divergence" in its current form for publication in Royal Society Open Science.

The comments of the reviewer(s) who reviewed your manuscript are included at the foot of this letter.

on behalf of Professor Matjaz Perc (Associate Editor) and Mark Chaplain (Subject Editor)
openscience@royalsociety.org

Associate Editor Comments to Author (Professor Matjaz Perc):

Thank you for the comprehensive revision of your manuscript, which we are happy to accept for publication in RSOS.

Appendix A

Comments on manuscript:

Replicator-mutator dynamics of linguistic convergence and divergence

by H. Kauhanen

This paper investigates the phenomenon of sociolinguistic identity maintenance using a multi-population extension of the replicator-mutator equation. The main assumption, which is well-established in sociolinguistics, is that people tend to align their use of language to the linguistic behaviour of their own ingroups and to simultaneously diverge from the language use of outgroups. The mathematical modelling of this phenomenon when it comes to language variation and change is understudied and this paper provides an interesting mathematical model.

Overall, the paper is well-written and well-structured. It is rather long, but the technicalities of the analysis of the model as well as the discussion of limitations fully justify the length. The mathematical model provided is appropriate for a first study of this phenomenon and the analysis of the dynamics of the system is convincing and seems sound. In addition, the drawbacks and limitations of the model are explicitly discussed and assessed. The writing style is suitable for a wide audience, despite the technicality of the method required for the stability analysis. In my opinion, any mathematically-versed reader should be able to understand the analysis. The illustration also help the reader to understand the dynamics of the system.

Regarding the state-of-the-art presentation, the author categorizes the model in two different classes, agent-based models and population model and they claim that their model falls in between these two extremes. While this is true, the boundary between agent-based models and population models is not as sharp as the author suggest. Take for example the Utterance Selection Model and derivative thereof cited by the authors in references [5,17,19,21,49]. This model admits a population limit that captures most of its dynamics and requires only 4 independent parameters, see Michaud (2017) for details. This paper also present a methodology to derive multi-population models from the stochastic agent-based dynamics and, therefore, drastically reduces the computational load of the model. However, it does not consider a population-wide distinction between group as the current paper does.

The author argues convincingly that variant not only have an intrinsic value but an extrinsic, socially-grounded value. This idea is not new and relates for example to the notion of prestige [19,42], momentum [17], or preference [21] for a variant. For instance, the model presented in [21] has been related to EGT in Michaud (2020). This short paper demonstrates

that the preferences for the available variants can be interpreted as a fitness qualified as “subjective” and corresponding to the extrinsic fitness mentioned here, while the “objective” fitness would correspond to the intrinsic fitness of a variant. The aim in Michaud (2020) and reference [21] was to show how the extrinsic fitness of a variant builds up, considering a single group and, thus, only convergence. But the discussion seems relevant in the light of the work presented here, especially on page 5, where the concept of intrinsic and extrinsic fitnesses are discussed.

The model presented is a multi-population replicator-mutator equation in which the fitness, qualified as “extrinsic”, is used to encode the convergence-divergence issue using a non-linear formulation that reflects the fact the fitness of a variant increases when it is used by the ingroups and when it is not used by the outgroups. While the two-population version of the model is in principle analytical (it involves solving two different cubic equations and combining them to find rest points. This always leads to 9 complex solutions and acceptable rest points must be real with both components bounded between 0 and 1), the authors restrict their analysis to the fully symmetric case, for which a phase portrait is provided. At the center of their analysis is the fact that 3 of the 4 phases are characterized by the existence of so-called “stable divergent states”.

The empirical application of the model relies on a single example (the backing of the STRUT vowel) and a qualitative application of the model to this test case. Choosing a single data set when more are available (footnote 7 on page 18 mentions at least two other data sets showing the same pattern) questions whether this example has been selected for convenience. Given the nature of the data, I would have liked to see the values for the other cases mentioned in footnote 7 on page 18. Can the mutation rate also be computed in this case? Are these data point also compatible with a qualitative application of the model?

Given the simple version of the model, I would have liked a quantitative evaluation of the model, or at least a discussion of it, rather than a qualitative one. For instance, for a value of μ calibrated from another source, it would have been straightforward to solve for σ given the observed displacement (denoted as $\delta_{\bar{D}}$ in appendix B and computed analytical on page 22, line 18) characterizing the $\mathbf{e}_m = (0.42, 0.58)$ (p.12, l.25). Or, given the observed displacement value of 0.08, it is possible to solve for the pair (σ, μ) compatible with this implicitly given by $\delta_{\bar{D}} = 0.08$. Doing a quick computation, it turns out that the quantitative fit is not possible. Given a mutation value of 0.23 (Eq. 13 on page 12), there are no value of σ for which the predicted displacement is 0.08. The minimum value is around 0.14. This means that there are no intersections between the curve $\delta_{\bar{D}} = 0.08$ and $\mu = 0.23$. The

minimum value for μ computed from $\delta_{\bar{D}} = 0.08$ is $\mu_{\min} \approx 0.2436$. This suggests that the calibration provided for μ underestimates the real value if the fully symmetric model is correct and suggests a rather small value for σ . I think that this analysis is missing from the paper and should be discussed. In particular, it suggests a rather weak social influence between the two groups investigated. For readability, it might be a good idea to add the curves $\delta_{\bar{D}} = 0.08$ and $\mu = 0.23$ to Figure 1 to illustrate this point.

The fully symmetric model suffers from a number of limitations and the authors tackled this by performing numerical simulations of a 3-population version of the model that includes the adult group as a driving force towards the expected equilibrium. One drawback of this model is the drastic increase in the number of parameters used (2 for the fully symmetric model and 6 for the 3-population model). This increase in parameter space makes mathematical analysis intractable and thus forces the use of numerical simulations to analyze the model. Once again, the evaluation of the model is purely qualitative and the criteria for validation is the almost sure convergence to a divergence stable state. This is a rather weak requirement and it turns out that the model often reaches such a state, especially when both μ and σ are high. As for the two-population model, the evaluation is purely qualitative and it would have been interesting to also shortly discuss the quantitative quality of the model. In addition, this version of the model includes an asymmetric mutation rate, which breaks the strategy symmetry assumed in the fully symmetric model.

I wonder whether it is possible for the authors to comment on the dynamics of the “almost” fully symmetric model, where the mutation rate between the variants is asymmetric and all the rest is as in the fully symmetric model. Whilst it is more complicated to analyze, this model is still analytically tractable and the resulting phase portrait can help the reader to grasp the effect of such an asymmetry.

Finally, in the discussion page 18 about extensions of the work, the estimate for the value of σ is discussed. In light of the quantitative application of the model discussed above, I am wondering the added value of this. For instance, the quantitative analysis already suggests that there are no values for σ compatible with the estimated mutation value μ , I expect this to stay true in the case of asymmetric mutation rate and even in the presence of adults. Could the author comment on this? If both a value for μ and σ are calibrated from data, then the displacement is predicted, but this will likely predict a higher displacement than the one observed.

References

- MICHAUD, J. (2017). Continuous time limits of the utterance selection model. *Phys. Rev. E*, **95**, 022308.
- MICHAUD, J. (2020). A game theoretic perspective on the utterance selection model for language change. In A. RAVIGNANI, C. BARBIERI, M. MARTINS, M. FLAHERTY, Y. JADOUL, E. LATTENKAMP, H. LITTLE, K. MUDD & T. VERHOEF, Eds., *The Evolution of Language: Proceedings of the 13th International Conference (EvoLang13)*.

Appendix B

Manuscript RSOS-201682

“Replicator–mutator dynamics of linguistic convergence and divergence”

Response to Reviewers

Below I respond point-by-point to the reviewers’ comments. In addition to changes prompted by those queries specifically, I have made the following stylistic changes to the manuscript:

- p. 4: “to divert their use of language” > “to distance their use of language”
- p. 11 (Table 1): “VC context” > “VC position” and “CV context” > “CV position”
- p. 15: “literature on ingroup favouritism” > “literature on social biases”
- p. 17: “tractability is almost surely lost” > “tractability is almost certainly lost”
- p. 20 (fn. 8): “Code availability” > “Data Accessibility”

To keep the paper’s length within reasonable limits after incorporation of new material prompted by review #3, I have cut one tangential paragraph from the Discussion section (beginning “Mathematical models are sometimes criticized...” in the original submission).

Review #1

This is an excellent paper. It draws on existing literature of evolutionary game-theoretic modeling of language dynamics, but it breaks considerable new ground. This applies both to the technical aspects (non-linear fitness function; explicit modeling of mutations) and the empirical coverage. The fact that the equilibrium analysis is linked with results from an empirical study from the literature is especially impressive.

Response: I thank the reviewer for their supportive comments.

Review #2

I found one tiny mistake: At the bottom of p21 line 58, I think you mean non-hyperbolic

Response: The referee is right, this should be “non-hyperbolic”.

Action: Typo corrected (p. 19 of the revised submission).

Review #3

This paper investigates the phenomenon of sociolinguistic identity maintenance using a multi-population extension of the replicator-mutator equation. The main assumption, which is well-established in sociolinguistics, is that people tend to align their use of language to the linguistic behaviour of their own ingroups and to simultaneously diverge from the language use of outgroups. The mathematical modelling of this phenomenon when it comes to language variation and change is understudied and this paper provides an interesting mathematical model.

Overall, the paper is well-written and well-structured. It is rather long, but the technicalities of the analysis of the model as well as the discussion of limitations fully justify the length. The mathematical model provided is appropriate for a first study of this phenomenon and the analysis of the dynamics of the system is convincing and seems sound. In addition, the drawbacks and limitations of the model are explicitly discussed and assessed. The writing style is suitable for a wide audience, despite the technicality of the method required for the stability analysis. In my opinion, any mathematically-versed reader should be able to understand the analysis. The illustration also help the reader to understand the dynamics of the system.

Response: I thank the referee for their positive assessment.

Regarding the state-of-the-art presentation, the author categorizes the model in two different classes, agent-based models and population model and they claim that their model falls in between these two extremes. While this is true, the boundary between agent-based models and population models is not as sharp as the author suggest. Take for example the Utterance Selection Model and derivative thereof cited by the authors in references [5,17,19,21,49]. This model admits a population limit that captures most of its dynamics and requires only 4 independent parameters, see Michaud (2017) for details. This paper also present a methodology to derive multi-population models from the stochastic agent-based dynamics and, therefore, drastically reduces the computational load of the model. However, it does not consider a population-wide distinction between group as the current paper does.

Response: I am grateful to the reviewer for bringing Michaud's (2017) work to my attention. On reflection, it is true that considerable progress can be made by considering continuous-time limits of full models, as Michaud (2017) demonstrates. I have added a qualification to this effect at the top of p. 2 of the revised submission, along with a reference to Michaud (2017).

Action: Parenthetical qualifier added at top of p. 2.

The author argues convincingly that variant not only have an intrinsic value but an extrinsic, socially-grounded value. This idea is not new and relates for example to the notion of prestige [19,42], momentum [17], or preference [21] for a variant. For instance, the model presented in [21] has been related to EGT in Michaud (2020). This short paper demonstrates that the preferences for the available variants can be interpreted as a fitness qualified as "subjective" and corresponding to the extrinsic fitness mentioned here, while the "objective" fitness would correspond to the intrinsic fitness of a variant. The aim in Michaud (2020) and reference [21] was to show how the extrinsic fitness of a variant builds up, considering a single group and, thus, only convergence. But the discussion seems relevant in the light of the work presented here, especially on page 5, where the concept of intrinsic and extrinsic fitnesses are discussed.

Response: Again, I am grateful to the referee for bringing this new work to my attention. The fact that a version of the utterance selection model can be interpreted in terms of replicator–mutator dynamics is certainly an interesting result. This has the potential for opening up a new research direction bridging the gap between models specified at the level of individuals (as done by Michaud) and models specified at the level of populations (as in the present work). I have added a note to this effect on p. 17 of the revised submission.

Action: Discussion added on p. 17.

The model presented is a multi-population replicator-mutator equation in which the fitness, qualified as "extrinsic", is used to encode the convergence-divergence issue using a non-linear formulation that reflects the fact the fitness of a variant increases when it is used by the ingroups

and when it is not used by the outgroups. While the two-population version of the model is in principle analytical (it involves solving two different cubic equations and combining them to find rest points. This always leads to 9 complex solutions and acceptable rest points must be real with both components bounded between 0 and 1), the authors restrict their analysis to the fully symmetric case, for which a phase portrait is provided. At the center of their analysis is the fact that 3 of the 4 phases are characterized by the existence of so-called “stable divergent states”. The empirical application of the model relies on a single example (the backing of the STRUT vowel) and a qualitative application of the model to this test case. Choosing a single data set when more are available (footnote 7 on page 18 mentions at least two other data sets showing the same pattern) questions whether this example has been selected for convenience. Given the nature of the data, I would have liked to see the values for the other cases mentioned in footnote 7 on page 18. Can the mutation rate also be computed in this case? Are these data point also compatible with a qualitative application of the model?

Response: The referee is right to express concern over the use of a single case study to validate a mathematical model. In response, I note that the primary concern of the present paper is to formulate the model and to explore its behaviour within the limits of analytical tractability, and only secondarily to test its predictions vis-à-vis empirical data. An extended empirical evaluation, given the complex nature of sociolinguistic data, necessarily falls beyond the scope of the paper, and will have to be carried out in a follow-up study.

Having said that, it is not an unreasonable request to take a more detailed look at the two other data points mentioned in footnote 7 of p. 18 of the original submission. Here I must admit to a previous confusion on my part, however. The two data points mentioned are reported in Figure 3 of Eckert & Labov (2017). I had originally read this figure as presenting the probabilities of backing of the DRESS vowel /ɛ/ and raising of PRICE /aɪ/. However, on a second reading, what is reported is in fact the factor weights from a VARBRUL (multinomial regression) analysis (Johnson 2009). These cannot be equated with usage probabilities (unless the VARBRUL input probability is equal to 0.5). Delving deeper into the literature, I have found that Eckert (2000, p. 112, Table 5.7) reports the relevant input probabilities alongside the factor weights. The input probabilities are 0.262 for DRESS backing and 0.009 for PRICE raising, and the factor weights are 0.540 for burnouts and 0.467 for jocks (DRESS) and 0.707 for burnouts and 0.257 (PRICE). Using the standard procedure for translating these weights into probabilities (Johnson 2009), we find that the probability of DRESS backing is 0.29 in burnouts and 0.24 in jocks, while the probabilities of PRICE raising are 0.02 (burnouts) and 0.003 (jocks). Thus, these data points are in fact rather different from the case of STRUT backing explored in the present paper, as the probabilities of use in the two social groups fall on the same side of 0.5, rather than on opposite sides. (For completeness, let it be noted that the input probability for STRUT backing given in Eckert (2000) is $0.494 \approx 0.5$, while the factor weights are 0.571 and 0.437 in burnouts and jocks, respectively. These translate to usage probabilities of 0.57 and 0.43, which are in line with the original data analysis presented in Eckert (1988) and cited in the manuscript, and correspond to a stable divergent state of the model.)

A reasonable generalization of the notion of “stable divergent state” to cases such as DRESS and PRICE here may exist, and the model may predict these data points for some combination of its parameter values. It is also possible that, empirically, DRESS and PRICE in fact correspond to stable convergent states, with the differences between jocks and burnouts explainable by sampling error. Exploring these complications in an empirically satisfactory way is beyond the remit of the present paper, however. I am grateful to the reviewer for bringing the complications to my attention, as they do suggest very fruitful avenues for future research. Correspondingly, I have added a brief note to this effect to the Discussion section, p. 16 of the revised submission.

Action: Footnote 7 of the original submission removed and its content incorporated and expanded, where applicable, to discussion on p. 16 (last sentence of first paragraph) of the revised submission.

Given the simple version of the model, I would have liked a quantitative evaluation of the model, or at least a discussion of it, rather than a qualitative one. For instance, for a value of μ calibrated from another source, it would have been straightforward to solve for σ given the observed displacement (denoted as $\delta_{\bar{D}}$ in appendix B and computed analytical on page 22, line 18) characterizing the $e_m = (0.42, 0.58)$ (p.12, l.25). Or, given the observed displacement value of 0.08, it is possible to solve for the pair (σ, μ) compatible with this implicitly given by $\delta_{\bar{D}} = 0.08$. Doing a quick computation, it turns out that the quantitative fit is not possible. Given a mutation value of 0.23 (Eq. 13 on page 12), there are no value of σ for which the predicted displacement is 0.08. The minimum value is around 0.14. This means that there are no intersections between the curve $\delta_{\bar{D}} = 0.08$ and $\mu = 0.23$. The minimum value for μ computed from $\delta_{\bar{D}} = 0.08$ is $\mu_{\min} \approx 0.2436$. This suggest that the calibration provided for μ underestimates the real value if the fully symmetric model is correct and suggest a rather small value for σ . I think that this analysis is missing from the paper and should be discussed. In particular, it suggest a rather weak social influence between the two groups investigated. For readability, it might be a good idea to add the curves $\delta_{\bar{D}} = 0.08$ and $\mu = 0.23$ to Figure 1 to illustrate this point.

Response: The referee is right to note that the modelled Eckert state $e_m = (0.42, 0.58)$ is not predicted, in an exact quantitative sense, by the model for the specific value of the mutation rate parameter μ estimated from the literature on phoneme confusions. I have the following remarks to make.

Firstly, the present study follows the precedent set by other evolutionary game theoretic work, particularly in biology but now also in linguistics (e.g. Baumann & Ritt 2017), in carrying out empirical evaluation on a qualitative level. While I am in principle sympathetic with the reviewer's concern over the limitations of this approach, and agree that more evaluation of a quantitative nature needs to be done in linguistics, I would argue that this is better suited for more complex models, not simple models like the present one, which are admittedly unable to take into account many of the complexities of the social dynamics of language. A quantitatively exact fit between model and data would be a rather unlikely expectation under these circumstances, as I had already stressed in the Discussion section of the original submission.

Secondly, it has to be stressed that an amount of uncertainty applies to the empirical estimates: both e_m and μ are point estimates, and the confidence intervals are unknown. As the reviewer notes, the Eckert state e_m is quantitatively predicted for a mutation rate approximately $\mu_{\min} \approx 0.2436$. This is not terribly far from our estimate $\mu = 0.23$.

To respond to the reviewer's query, I have added a new paragraph on pp. 15–16 of the revised manuscript, together with a new figure (Figure 3, p. 16) that illustrates how displacement δ relates to mutation rate μ and social alignment σ . This figure makes it explicit that the exact frequencies of the Eckert state e_m are not attested (in an exact quantitative sense) for the mutation rate estimate assumed in the paper. It also makes it clear that a quantitative fit is possible for near values of μ (such as $\mu = 0.25$). This corresponds to a value of σ of about $\sigma = 0.1$ (as the reviewer notes, the social influence between the two groups is predicted to be weak). This prediction is in line with empirical studies which have found that ingroup favouritism in humans is likely to result from a pro-ingroup rather than an anti-outgroup bias (Balliet et al. 2014; Ref. [63] in the manuscript).

Action: Discussion of quantitative model evaluation added on pp. 15–16, along with new Figure 3 (p. 16).

The fully symmetric model suffers from a number of limitations and the authors tackled this by performing numerical simulations of a 3-population version of the model that includes the adult group as a driving force towards to expected equilibrium. One drawback of this model is the drastic increase in the number of parameters used (2 for the fully symmetric model and 6 for the 3-population model). This increase in parameter space makes mathematical analysis intractable and

thus forces the use of numerical simulations to analyze the model. Once again, the evaluation of the model is purely qualitative and the criteria for validation is the almost sure convergence to a divergence stable state. This is a rather weak requirement and it turns out that the model often reaches such a state, especially when both μ and σ are high. As for the two-population model, the evaluation is purely qualitative and it would have been interesting to also shortly discuss the quantitative quality of the model. In addition, this version of the model includes an asymmetric mutation rate, which breaks the strategy symmetry assumed in the fully symmetric model.

I wonder whether it is possible for the authors to comment on the dynamics of the “almost” fully symmetric model, where the mutation rate between the variants is asymmetric and all the rest is as in the fully symmetric model. Whilst it is more complicated to analyze, this model is still analytically tractable and the resulting phase portrait can help the reader to grasp the effect of such an asymmetry.

Response: I agree with the reviewer that an analytical exploration of certain weakly generalized versions of the fully symmetric game is an exciting prospect; unfortunately, I have been unable to solve the relaxed model the reviewer has in mind here. I have added discussion to this effect on p. 17 of the revised submission.

Action: Discussion added on p. 17.

Finally, in the discussion page 18 about extensions of the work, the estimate for the value of σ is discussed. In light of the quantitative application of the model discussed above, I am wondering the added value of this. For instance, the quantitative analysis already suggests that there are no values for σ compatible with the estimated mutation value μ , I expect this to stay true in the case of asymmetric mutation rate and even in the presence of adults. Could the author comment on this? If both a value for μ and σ are calibrated from data, then the displacement is predicted, but this will likely predict a higher displacement than the one observed.

Response: As I have explained above, I consider a qualitative evaluation most appropriate at the present time. While the reviewer is correct to point out the importance of a quantitative approach, and is probably correct about the particular predictions above, exploration of these will have to be left to a follow-up study.

The added value of mentioning possibilities of estimating the strength of the social alignment parameter σ (or the entire matrices in asymmetric cases) is that this connects the present modelling paradigm with an existing empirical literature on biases in the social dynamics of human populations, as reviewed for instance in Balliet et al. (2014) (Ref. [63]). As pointed out above, this literature predicts that the value of the social alignment parameter σ ought to be rather small, and this accords well with the new quantitative evaluation of the model prompted by the reviewer’s comments. This connection is now stated explicitly on p. 16 of the revised submission.

Action: Connection to empirical literature on social biases clarified in the new paragraph on p. 16.

References

Balliet D, Wu J, De Dreu CKW. 2014 Ingroup favoritism in cooperation: a meta-analysis. *Psychological Bulletin* **140**, 1556–1581.

Baumann A, Ritt N. 2017 On the replicator dynamics of lexical stress: Accounting for stress-pattern diversity in terms of evolutionary game theory. *Phonology* **34**, 439–471.

- Cutler A, Weber A, Smits R, Cooper N. 2004 Patterns of English phoneme confusions by native and non-native listeners. *The Journal of the Acoustical Society of America* **116**, 3668–3678.
- Eckert P. 1988 Adolescent social structure and the spread of linguistic change. *Language in Society* **17**, 183–207.
- Eckert P. 2000 *Linguistic variation as social practice*. Malden, MA: Blackwell.
- Eckert P, Labov W. 2017 Phonetics, phonology and social meaning. *Journal of Sociolinguistics* **21**, 467–496.
- Johnson, D. E. 2009 Getting off the GoldVarb standard: introducing Rbrul for mixed-effects variable rule analysis. *Language and Linguistics Compass* **3**, 359–383.
- Michaud J. 2017 Continuous time limits of the utterance selection model. *Physical Review E* **95**, 022308.
- Michaud J. 2020 A game theoretic perspective on the utterance selection model for language change. In Ravignani A, Barbieri C, Martins M, Flaherty M, Jadoul Y, Lattenkamp E, Little H, Mudd K, Verhoef T, editors, *The Evolution of Language: Proceedings of the 13th International Conference (EvoLang13)*.